

# Direct O₂ control on the partitioning between denitrification and dissimilatory nitrate reduction to ammonium in lake sediments

Adeline N.Y. Cojean[1], Jakob Zopfi[1], Alan Gerster[1], Claudia Frey[1], Fabio Lepori[2], Moritz F. Lehmann[1]

[1]Department of Aquatic and Stable Isotope Biogeochemistry, University of Basel,
Bernoullistrasse 30, CH-4056 Basel
[2]Institute of Earth Science, Scuola Universitaria Professionale della Svizzera Italiana (SUPSI), Trevano, CH-6952 Canobbio, Switzerland

*Correspondence to*: Adeline N.Y. Cojean (adeline.cojean@unibas.ch)

**Abstract.** Lacustrine sediments are important sites of fixed nitrogen (N) elimination through the reduction of nitrate to N₂ by
denitrifying bacteria, and are thus critical for the mitigation of anthropogenic loading of fixed N in lakes. In contrast, dissimilatory nitrate reduction to ammonium (DNRA) retains bioavailable N within the system, promoting internal eutrophication. Both processes are thought to occur under oxygen-depleted conditions, but the exact O₂ thresholds particularly of DNRA inhibition are uncertain. In O₂-manipulation laboratory experiments with dilute sediment slurries and $^{15}NO_3^-$ additions at low- to sub-micromolar O₂ levels, we investigated how, and to what extent, oxygen controls the balance
between DNRA and denitrification in lake sediments. In all O₂-amended treatments, oxygen significantly inhibited both denitrification and DNRA compared to anoxic controls, but even at relatively high O₂ concentrations ($\geq 70$ µmol L$^{-1}$), nitrate reduction by both denitrification and DNRA was observed, suggesting a relatively high O₂ tolerance. Nevertheless, differential O₂ control and inhibition effects were observed for denitrification versus DNRA in the sediment slurries. Below 1 µmol L$^{-1}$ O₂, denitrification was favored over DNRA, while DNRA was systematically more important than denitrification
at higher O₂ levels. Our results thus demonstrate that O₂ is an important regulator of the partitioning between N-loss and N-recycling in sediments. In natural environments, where O₂ concentrations change in near bottom waters on an annual scale (e.g., overturning lakes with seasonal anoxia), a marked seasonality with regards to internal N eutrophication versus efficient benthic fixed N elimination can be expected.

## 1 Introduction

Over the last decades, intensive human activities have dramatically affected the nitrogen (N) cycle in aquatic systems through elevated inputs of reactive (biologically available) N. In some lakes, external N loading can lead to excessive algal blooms in the upper water column, and the subsequent decomposition of the sinking algal biomass is often associated with O₂ depletion in the deeper water column, and possibly, seasonal or permanent anoxia (e.g. Blees et al., 2014; Lehmann et al., 2004, 2015). Depending on the O₂-concentrations in the water column, and the reactivity of the sediment
organic matter, the oxygen penetration depth within lacustrine sediments can vary (e.g. Lehmann et al., 2009), and so will



the transition zone that separates aerobic from anaerobic biogeochemical reactions. Under oxygen-depleted conditions, both in the water column and in sediments, anaerobic N-transformation processes such as denitrification, anammox and/or dissimilatory nitrate reduction to ammonium (DNRA) become important. While denitrification and anammox can mitigate excessive N loading (eutrophication) by converting reactive nitrogen ($NO_3^-$, $NO_2^-$, $NH_4^+$) to $N_2$, which subsequently returns to

the atmosphere, DNRA retains a bioavailable form of nitrogen within the ecosystem, fostering internal eutrophication of lakes (Tiedje, 1988).

The biogeochemical conditions that regulate the partitioning between these different N-transforming processes (and others) in benthic environments remain uncertain, but links to other biogeochemical cycles are likely an important factor. It

is commonly accepted, that when organic matter and nitrate concentrations are relatively high, nitrate is mostly reduced to $N_2$ by denitrifying bacteria (Gruber, 2008; Seitzinger et al., 2006; Seitzinger, 1988). In contrast, DNRA seems to be favored in sediments with an excess of electron donors (total organic carbon (TOC), $H_2S$, $Fe^{2+}$) relative to nitrate (Brunet and Garcia-Gil, 1996; Roberts et al., 2014). In organic matter-rich lake sediments, the contribution of anammox to the total fixed-N transformation fluxes across sediment-water interfaces is most likely minor relative to both denitrification and DNRA, since

anammox seems to occur primarily in sediments with low organic matter content (Babbin et al., 2014; Thamdrup and Dalsgaard, 2002).

Changes in the redox zonation may have profound impact on the benthic N cycle (Otte et al., 1996). Seasonal cycles of water column mixing and stagnation can modulate the penetration of redox boundaries into the sediments, potentially

changing the redox environments of, for example, nitrifiers, denitrifiers, anammox, and DNRA bacteria. Oxygen can hence be considered a key regulator of benthic N exchange (Glud, 2008; Tiedje, 1988), and its effects are multifold. On one side, increasing $O_2$ concentrations may expand the oxic/suboxic zone where nitrification can supply $NO_3^-$ and $NO_2^-$ for denitrification and anammox, enhancing the overall fixed-N loss (Lehmann et al. 2015). On the other hand, $O_2$ can inhibit nitrate/nitrite reduction. When surface sediments are oxygenated, the facultative anaerobic microbes will preferably use

oxygen, as the heterotrophic respiration with $O_2$ yields more energy to cells for growth than with other oxidants (i.e., $NO_3^-$, $NO_2^-$; Froelich et al., 1979; Payne et al., 2009; Thauer et al., 1977). Moreover, under oxygenated conditions, the synthesis and/or the activity of the key enzymes involved in nitrate/nitrite reduction may be suppressed (Körner and Zumft 1989, Baumann et al. 1996, Dalsgaard et al., 2014). Existing reports on $O_2$ tolerance and inhibition of denitrification and anammox in environments differ quite significantly. Inhibition may occur already at very low (nanomolar) levels of $O_2$ (Dalsgaard et

al., 2014), but experimental studies also revealed that relatively high $O_2$ levels may be required (up 16% saturation levels) to induce a 50% inhibition of anammox (Jensen et al., 2008; Kalvelage et al., 2011). The apparent persistence of denitrification at relatively high $O_2$ concentration levels led to a revision of the classical paradigm regarding the absolute $O_2$ inhibition of nitrate reduction in nature (Tiedje et al., 1988), with important implications regarding the total volume of hypoxic zones in the ocean or in lakes that hosts microbial $N_2$ production (Paulmier and Ruiz-Pino, 2009).






While oxygen inhibition/tolerance of denitrification and anammox has been studied previously in the ocean water column (Jensen et al. 2008, Kalvelage et al. 2011, Babbin et al. 2014, Dalsgaard et al. 2014), investigations into the $O_2$ control on benthic N-reduction are rather rare, and limited to sandy and low organic matter marine sediments (Gao et al., 2010; Jäntti and Hietanen, 2012; Rao et al., 2007). Despite intensified research, the exact $O_2$ thresholds with regards to the

direct inhibition of benthic N reduction are still poorly constrained. This is particularly true for DNRA. Recent work has highlighted the significance of DNRA even in the presence of relatively high $O_2$ concentrations (i.e., at hypoxic levels (i.e., 10-62 mmol/L), or concentrations even >62 mmol/L) in estuarine sediments (Roberts et al., 2012, 2014) and marine sediments (Jäntti and Hietanen, 2012), but a systematic investigation of how DNRA is impacted at low micromolar $O_2$ levels in aquatic sediments (and how in turn the balance between denitrification and DNRA is affected), does, to our knowledge,

not exist.

In this study, we provide first experimental evidence for direct $O_2$ control on the fate of reactive N in lacustrine sediments with high organic matter content. Through slurry incubation experiments with sediment from a eutrophic lake in Switzerland (Lake Lugano), $^{15}$N-labelled substrates and manipulated $O_2$ concentrations, we investigated the functional

response of benthic N-reducing processes to changing $O_2$ levels. We demonstrate that denitrification and DNRA are differentially sensitive towards $O_2$, which has important implications for fixed N removal in environments that undergo short- and longer-term $O_2$ changes, such as seasonally stratified (anoxic) lakes or other aquatic environments with expanding volumes of hypoxia/anoxia.

## 2 Sampling site, materials and methods

### 2.1 Sampling location

Sediment sampling took place in the south basin of Lake Lugano, a natural alpine lake situated at the border between Switzerland and Italy. Between April and January, the water column of the basin is stratified, with bottom-water suboxia/anoxia starting in late spring/early summer (e.g., Lehmann et al. 2004; Lehmann et al. 2015). During winter (February/March) the lake turns over and bottom waters are ventilated until the water column re-stratifies in spring, and

bottom-water $O_2$ concentrations decrease again (Fig. 1). Water column $O_2$ and N-compound ($NO_3^-$, $NO_2^-$, $NH_4^+$) concentrations were measured as part of a long-term monitoring campaign promoted by the International commission for the protection of Italian-Swiss waters (CIPAIS; Commissione Internazionale per la Protezione delle Acque Italiano-Svizzere) and run by SUPSI (University of Applied Sciences and Arts of Southern Switzerland) on behalf of the Admnistration of Canton Ticino. Sediment cores were taken at two locations, Figino (8°53'37"E, 45°57'31"N, 94 m depth) and Melide

(8°57'29"E, 45°56'22"N, 85 m depth) in October 2017, using a small gravity corer (inner diameter 6.2 cm). Figure 1 displays representative seasonal trends in the deep south basin. During oxygenation of the bottom waters, nitrate





concentrations in the water 2 m above the sediments reach about 75 µmol $L^{-1}$, and even during water column anoxia, near-sediment nitrate concentrations rarely dropped below 15 and 5 µmol $L^{-1}$ at Figino and Melide, respectively (Fig. 1; SUPSI data), so that the sediments are constantly exposed to nitrate-containing bottom waters. Ammonium concentrations in bottom

water were relatively high (~ 30-140 µmol $L^{-1}$) during anoxia and close to the detection limit during months when the water column was mixed.

### 2.2 Porewater sampling

Porewater oxygen microprofiles were generated using an $O_2$ microsensor (Unisense) with a tip diameter of 100 µm in duplicate cores from both sites. The overlying water was gently stirred (without disturbing the sediment water interface)

and aerated to determine the $O_2$ penetration depth under oxygenated conditions. Porewater samples for the analysis of dissolved inorganic nitrogen concentrations were obtained by sectioning of a separate set of cores from the same sites at 1 cm-interval, and centrifuging of the samples.

### 2.3 N-transformation incubation experiments

In a first step, incubations to measure potential denitrification and DNRA rates under control (i.e., anoxic)

conditions were performed. At each site, fresh surface sediments (upper 2 cm) from duplicate sediment cores were homogenized to prepare dilute sediment slurries. Aliquots of 1 g sediment and 70 mL anoxic artificial lake water ($NO_3^-$, $NO_2^-$, $NH_4^+$-free) were transferred into 120 mL serum bottles. The use of dissolved-$NO_x$-free artificial water is important to avoid any potential underestimation of N-transformation process rates due to $^{28}N_2$ production from ambient $NO_3^-$ or $NO_2^-$-present in bottom waters. Serum bottles were sealed and crimped using blue butyl rubber stoppers and aluminum caps. The

sediment slurries (generally in triplicates, Table 1) were He-flushed for 10 min to lower the atmospheric $N_2$ and $O_2$ backgrounds, and placed overnight on a shaker (80 rpm) at 8 °C in the dark to remove any residual $O_2$. Labeled $^{15}N$ substrate (i.e., $Na^{15}NO_3^-$, 99% $^{15}N$, Cambridge Isotopes Laboratories) was added in order to quantify potential rates of denitrification and DNRA. During the incubation period (ca. 10 hours), anoxic sediment slurries were kept in an incubator on an orbital shaker (80 rpm; 8°C). For subsampling of gas and liquid, the incubation vials were transferred to an anaerobic chamber with

$N_2$-atmosphere. There, two-milliliter gas samples were taken at four successive time points ($T_0, T_1, T_2, T_3$) for $N_2$ isotope measurement, in exchange with 2 mL He ($T_0$) or anoxic Milli-Q water ($T_1$ to $T_3$) in order to compensate for any pressure decrease inside the vials. Gas samples were then transferred into 3 mL exetainers (Labco), prefilled with anoxic water, and stored upside down until isotope analysis. Liquid samples (6 mL) were taken at $T_0$ and $T_3$ for the quantification of DNRA rates through $N-NH_4^+$ isotope analysis (see below) and for the assessment of nutrient ($NO_3^-$, $NO_2^-$, $NH_4^+$) concentrations.

### 125 2.4 $O_2$ manipulation experiments

For the $O_2$ manipulation experiments, serum bottles were equipped with TRACE Oxygen Sensor Spots (TROXSP5, detection limit = 6 nmol $L^{-1}$ $O_2$, Pyroscience, Germany), allowing non-invasive, contactless monitoring of dissolved $O_2$





concentrations in the dilute slurry. The sensor spots were fixed at the inner side of the glass wall with silicone glue and the sensor signal was read out from outside using a Piccolo2 fiber-optic oxygen meter (PyroScience). Different volumes of pure

$O_2$ (99,995%) were injected into the headspace of pre-conditioned and $^{15}NO_3^-$-amended slurries using a glass syringe (Hamilton). For each treatment, the gas volume required to reach the targeted $O_2$ equilibrium concentration (0.8, 1.2, 2, …, 78.6 µmol $L^{-1}$) was calculated based on the headspace versus slurry volumes, salinity, and temperature (Gordon and Garcia, 1992). Measured $O_2$ concentrations in slurries after injection of the respective $O_2$ gas volumes were always close to the ones calculated (the first measurement was performed 30 minutes after injection to ensure gas equilibration between the gas and

the liquid phase). Oxygen concentrations in the slurry incubations were monitored with the fiber-optic oxygen meter every 30 minutes and, in case of a marked decline in dissolved $O_2$ due to microbial consumption, $O_2$ was added in order to return to the initial target $O_2$ concentrations. In addition to continuous agitation on the shaking table, the dilute slurries were vigorously shaken by hand every 30 minutes to avoid the formation of anoxic microniches, which may act to increase rates of anaerobic N-transformation processes (Kalvelage et al., 2011).

**2.5 Hydrochemical analyses**

Nitrite concentrations were determined colorimetrically using sulfanilamide and Griess-reagent, according to (Hansen and Koroleff 1999). Total $NO_x$ (i.e., $NO_3^-$ + $NO_2^-$) concentrations were measured using a NOx-analyzer (Antek Model 745), by reduction to nitric oxide (NO) in an acidic $V^{3+}$ solution, and quantification of the generated NO by chemiluminescence detection (Braman and Hendrix, 1989). Nitrate concentrations were then calculated from the difference

between $NO_x$ and $NO_2^-$ concentrations. Ammonium was measured by suppression-ion chromatography with conductivity detection (940 Professional IC Vario, Metrohm, Switzerland).

**2.6 $^{15}N$-based rate measurements**

For the determination of denitrification rates, gas samples from the $^{15}N$-isotope enrichment experiments were analyzed for $^{15}N/^{14}N$ isotope ratios in the $N_2$ using a Delta V Advantage isotope ratio mass spectrometer (IRMS; Thermo

Fisher Scientific) with a ConFlo IV interface (Thermo Fisher Scientific). Denitrification (and negligible anammox) rates were calculated based on the quantification of $^{15}N$ in the $N_2$ gas in excess of the natural abundance, i.e. by assessing the linear increase in the concentrations of $^{30}N_2$ (and to a minor extent $^{29}N_2$) with incubation time (Nielsen, 1992; Thamdrup and Dalsgaard, 2002). DNRA rates were quantified using the isotope-pairing method described by Risgaard-Petersen et al. (1995). Briefly, 2 mL liquid samples were transferred into 6 mL exetainers (Labco) and closed with plastic screw septum

caps. The headspace was flushed with He for 2 min to reduce the $^{28}N_2$ background, and 25 µL $mL^{-1}$ of alkaline (16 mol NaOH) hypobromite iodine solution (3.3 mol $L^{-1}$) were added through the septum to convert $NH_4^+$ to $N_2$ (Robertson et al., 2016). Exetainers were stored upside down and placed on a shaker (100 rpm) for 24 h at room temperature. The produced $N_2$ was then analyzed by IRMS as described above. DNRA rates were determined based on the $^{15}NH_4^+$ production with time, as




calculated from the total $^{15}$N-$N_2$ in the hypobromite-treated samples (i.e., calculated from the excess $^{29}N_2$ and $^{30}N_2$ signals). The recovery of $^{15}$N-label from identically treated standards was >95%.

**2.7 Statistics**

Results are presented as the mean and standard error of n replicate experiments (Table 1). Significance differences between results were verified using a Student's $t$-test (P <0.05) in Microsoft Excel software.

**3 Results**

**3.1 Porewater hydrochemistry**

The $O_2$ microsensor profiles revealed that the $O_2$ penetration at the two sites under aerated conditions ranged between 2.4 mm (Melide) and 3.7 mm (Figino, Fig. 2). The relatively low oxygen penetration depth is consistent with a high organic carbon content (~8%, data not shown). According to the observed $O_2$ concentration gradients at the two stations, the potential $O_2$ flux into sediments was greater at Melide suggesting a higher reactivity of the sedimentary organic matter. In contrast to the microsensor profiling, the sectioning-centrifuging technique was not sufficient to resolve the exact porewater nitrate concentration gradient, yet the observed nitrate concentration data across the sediment-water interface (Fig. 2) clearly indicate that the sediments at both sites represent a sink for water-column nitrate, and that nitrate is consumed to completion already within the top centimeter of the sediments. In contrast, ammonium concentrations just below the sediment-water interface at Figino and Melide increased steeply from 830 and 600 $\mu$mol L$^{-1}$ NH$_4^+$ to 1.7 and 1.2 mmol L$^{-1}$, respectively.

**3.2 N-transformations in control experiments**

Potential rates of denitrification and DNRA under true anoxic conditions were quantified at both sampling sites in October 2017. Anammox rates were measured in a previous study at different times of the year, and its contribution to the total fixed-N removal was always less than 1%, thus negligible with respect to other processes (Cojean, in prep.). Indeed, in all experiments, denitrification and DNRA were the main benthic N-transformation processes with an essentially equal contribution to the total nitrate reduction (≈0.1 $\mu$mol N g$^{-1}$ wet sed. d$^{-1}$; Table 1 caption). We ensured that measured DNRA rates were not underestimated due to $^{15}$NH$_4^+$ loss through adsorption on mineral surfaces. Previous results (Cojean, in prep.) demonstrate that adsorption of ambient or tracer ammonium does not occur at detectable levels in the dilute sediment slurries. Ammonium consumption by nitrifiers in presence of $O_2$, however, might affect quantification of DNRA rates. Indeed, oxic slurry incubation experiments (≥ 73 $\mu$mol L$^{-1}$ $O_2$) revealed that at least at high $O_2$ concentrations net NO$_3^-$ production occurs (≤ 1 $\mu$mol N g$^{-1}$ wet sed. d$^{-1}$). Thanks to the comparatively large background pool of ammonium (~ 42 $\mu$mol L$^{-1}$ derived from the porewater of the wet sediment and from organic matter N remineralization), the little $^{15}$NH$_4^+$





generated by DNRA during the incubation ($< 0.1$ μmol L$^{-1}$) is strongly diluted, preventing that a significant fraction of the DNRA-derived $^{15}NH_4^-$ is consumed by nitrifying microorganisms.

### 3.3 Impact of $O_2$ on $NO_3^-$ reduction in sediments

The $O_2$ sensitivity of denitrification and DNRA and inhibition kinetics were investigated through slurry incubation experiments under $O_2$-controlled conditions. At both sites, potential denitrification and DNRA rates consistently decreased with increasing $O_2$ concentration (Fig. 3). While the general pattern was systematic for both processes (i.e., an exponential attenuation of both denitrification and DNRA rates with increasing $O_2$), the response of denitrifiers versus nitrate ammonifiers towards manipulated $O_2$ differed across sites and treatments. We compared $O_2$-addition experiments to the

anoxic controls to estimate the inhibition of nitrate reduction by $O_2$. At the lowest $O_2$ concentration (~ 1 μmol L$^{-1}$ $O_2$), denitrification was less inhibited than DNRA at Figino (29% and 51% inhibition, respectively) while the suppression was almost equivalent at Melide (43% and 37% inhibition of denitrification and DNRA respectively, Table 1). At $O_2$ concentrations around $2 \pm 0.2$ μmol L$^{-1}$, both denitrification and DNRA rates were more than 50% inhibited compared to the anoxic control (Table 1, Fig. 3). At $O_2$ concentration $\geq 2$ μmol L$^{-1}$, DNRA rates were generally higher than those of

denitrification (with one exception, i.e., 16 μmol L$^{-1}$ $O_2$ at Figino; Fig. 3), indicating that denitrification was more sensitive than DNRA to elevated $O_2$ levels. Oxygen concentrations higher than 73 μmol L$^{-1}$ resulted in almost complete inhibition of denitrification at both sites (96% and 93% at Figino and Melide, respectively, Table 1). Oxygen inhibition thresholds for DNRA were even higher, as DNRA rates were significantly less impaired than denitrification at these elevated $O_2$ levels (79% and 75% inhibition compared to the anoxic controls at Figino and Melide, respectively; Table 1). Hence, the relative

contribution of the two processes to total nitrate reduction was significantly affected by changing $O_2$ concentrations. At anoxic and submicromolar levels of $O_2$ ($\leq 1 \pm 0.2$ μmol L$^{-1}$ $O_2$), denitrification rates were higher than those of DNRA, while at higher $O_2$ concentration the ratio between denitrification and DNRA was shifted in favor of nitrate ammonifiers (Fig. 4).

Consistent with the observed decline in denitrification and DNRA rates based on the $^{15}N_2$ and $^{15}NH_4^+$ measurements

in the $^{15}N$-label incubations, nitrate consumption in slurries decreased with increasing $O_2$ concentration at both stations (Table 1). Similarly, maximum ammonium accumulation was observed in the anoxic controls, whereas at higher $O_2$ levels ammonium underwent net consumption, indicating the concomitant decrease of DNRA and the increasing importance of nitrification under more oxic conditions, particularly at Melide. In incubations where nitrate concentrations decreased, the ratio of $(NO_3^-)_{reduced}:(N-N_2 + {}^{15}NH_4^+)_{produced}$ was always significantly higher (>5:1) than expected (1:1). This observation is

consistent with previous work in the Lake Lugano South Basin (Wenk et al., 2014). Here, whole-core flow-through incubations also revealed that $NO_3^-$ fluxes into the sediments significantly exceeded benthic $N_2$ production, an imbalance, which could neither be explained by nitrate reduction to ammonium nor incomplete reduction to $N_2O$. As it is not the scope of this study, we will not discuss this puzzling discrepancy further, but we speculate that excess $NO_3^-$ consumption may be linked to bacterial and algal uptake (Bowles et al., 2012). Biotic immobilization of $NO_3^-$ in marine sediments has been





attributed previously to the intracellular storage of nitrate by filamentous bacteria (Prokopenko et al., 2013; Zopfi et al., 2001) and/or diatoms (Kamp et al., 2011), but we do not know whether such nitrate sinks are important also in Lake Lugano sediments.

## 4 Discussion

### 4.1 Anaerobic N-cycling in the South Basin of Lake Lugano

Benthic denitrification and DNRA were the predominant anaerobic N-transformation processes at the two studied stations. Interestingly, the contribution of DNRA was systematically higher than observed in flow-through whole-core incubations performed with sediment from the same basin. Wenk et al. (2014) reported a maximum DNRA contribution to $NO_3^-$ reduction of not more than 12%, but also argued that their DNRA rate measurements must be considered conservative, because they did not account for the production of $^{14}NH_4^+$ from ambient natural-abundance nitrate. The reason for such a 230 discrepancy is unclear, but there seems to be a tendency for slurry incubations to yield higher DNRA rates compared to denitrification (Kaspar, 1983), implying biasing methodological effects. The observed discrepancies may also be related to natural sediment heterogeneity and/or seasonal/interannual fluctuations in benthic N transformation rates. As for the latter, in 2016, the annual water overturn and bottom-water ventilation was exceptionally suspended and sediments remained anoxic for more than a year. In contrast, in 2017, the water column mixed in January and surface sediments were oxygenated 235 throughout June. Our $O_2$ manipulation experiments revealed that redox conditions have a marked impact on the partitioning between the two nitrate reduction pathways, and consistent with the slurry incubation data, the extended $O_2$ exposure of microbes at the sediment-water interface in 2017 compared to the preceding year may have favored nitrate ammonifiers over denitrifiers. Independent of any possible spatio-temporal variability, in this study, DNRA rates were equal, or even higher, than denitrification. Such a partitioning of the two nitrate reducing processes is not implausible and was similarly observed 240 in a wide range of environments, particularly in more reduced sediments with high organic matter content and comparatively low nitrate levels (Brunet and Garcia-Gil, 1996; Dong et al., 2011; Papaspyrou et al., 2014). More generally, substrate-availability changes induced by $O_2$ fluctuations may be important drivers of the partitioning between denitrification and DNRA (Cojean in prep.), and environmental conditions that favor DNRA over denitrification may be quite common. However, to our knowledge, experimental evidence for the direct $O_2$ control on the balance between these two nitrate-245 reducing processes is still lacking.

### 4.2 $O_2$ inhibition thresholds of benthic nitrate reduction

Our study shows that submicromolar $O_2$ levels significantly lowered both, denitrification and DNRA rates. Denitrification and DNRA were inhibited by about 30-50% at 1 µmol $L^{-1}$ $O_2$, while in previous studies that investigated $O_2$ effects on fixed-N elimination in the water column, denitrification was almost completely suppressed at this $O_2$ level already. 250 For example, by conducting incubation experiments using samples from oxygen minimum zones in the Eastern Tropical





Pacific, a 50% inhibition of denitrification was noticed already at 0.2 µmol L$^{-1}$ O$_2$, and complete suppression at 1.5-3 µmol L$^{-1}$ O$_2$ (Dalsgaard et al., 2014, Babbin et al., 2014). Similarly, incubation experiments with samples from a Danish fjord exhibited full inhibition of denitrification at 8-15 µmol L$^{-1}$ O$_2$ (Jensen et al., 2009). In marine sediments, in contrast, denitrification was occurring even at O$_2$ concentrations greater than 60 µmol L$^{-1}$ (Gao et al., 2010, Rao et al., 2007). This is

in agreement with our results showing that at higher O$_2$ levels ($\geq$73 µmol L$^{-1}$) denitrification was still active although at very low rates compared to the anoxic control ($\geq$ 93% inhibition). Similarly, DNRA was still occurring, and was less impaired by the elevated O$_2$ concentration compared to denitrification ($\geq$ 75% inhibition relative to the anoxic control). An increase of DNRA relative to denitrification rates under oxic conditions (> 100 µmol L$^{-1}$ O$_2$) was also observed in estuarine sediments, though N-removal remained predominant (Roberts et al., 2012, 2014). In brackish sediments in the Gulf of Finland in the

Baltic Sea, at elevated O$_2$ concentrations (from 50 to 110 µmol L$^{-1}$ in bottom waters), benthic DNRA rates were generally higher than denitrification rates (Jäntti and Hietanen, 2012), further supporting our findings. Yet, in contrast to our study, their observations suggest a higher O$_2$ sensitivity (i.e., greater inhibition) of DNRA compared to denitrification in sediments with higher bottom water O$_2$ concentrations (> 110 µmol L$^{-1}$). Given the paucity and discrepancy of existing data in this context, it is premature to conclude that DNRA microbes are generally less or more oxygen-tolerant than denitrifiers. A

direct comparison of DNRA O$_2$ inhibition thresholds in this study and in the study of Jäntti and Hietanen (2012) is difficult because of the differing methodological approaches. There, nitrate reduction rates were determined in whole-core incubations, without manipulating (and measuring) the O$_2$ concentrations at the sediment depth where nitrate is actually reduced. And although the O$_2$ penetration depth and porewater O$_2$ concentrations will respond to a certain degree to the O$_2$ content in the bottom water, deducing the actual O$_2$ concentrations for the active nitrate reduction zone within the sediment

from O$_2$ concentrations in the overlying water is problematic. Here, we tested the oxygen sensitivity of a microbial community in suspension, directly exposed to defined O$_2$ conditions. These incubation data indicate that DNRA is less inhibited than denitrification at O$_2$ concentrations $\geq$ 73 µmol L$^{-1}$ and, at the same time, imply that anoxia per se is not a strict requirement for DNRA, as previous ecosystem-scale work has also suggested (Burgin and Hamilton, 2007). Our results also are consistent with observations made in soil microcosms showing that DNRA is less sensitive to increasing O$_2$ partial

pressures than denitrification within the range of 0-2% O$_2$ v/v (Fazzolari et al., 1998; Morley and Baggs, 2010).

The observed O$_2$ inhibition thresholds for nitrate reduction are significantly higher than reported from most incubation studies with water column samples (Dalsgaard et al., 2014, Babbin et al., 2014, Jensen et al., 2008). Elevated O$_2$ tolerance in prior studies was often attributed to the formation of anoxic microniches that may foster anaerobic N-reduction

(Kalvelage et al., 2011). It is unlikely that such microniches formed during our incubation experiments since slurries were heavily diluted (1 g sediment in 70 mL water) and vigorously shaken by hand every 30 min, in addition to the continuous agitation on a shaking table during the incubation. Also, experiments were replicated 2-3 times for some O$_2$-amended treatments, and measured rates were very similar between replicates. If anoxic microniches had formed, we would have



expected that their formation is more variable, resulting in a lower reproducibility of the determined rates. One might
speculate that benthic nitrate-reducing microbial communities are more tolerant towards higher $O_2$ levels than pelagic ones.

The existence of aerobic denitrifiers (e.g. microbes that reduce $NO_3^-/NO_2^-$ to $N_2$ in presence of $O_2$) in soils and
sediments has been confirmed through isolation of bacterial strains (e.g. Robertson et al., 1995), and it was suggested that
they contribute to the total fixed N loss in marine sediments (Carter et al., 1995; Patureau et al., 2000; Zehr and Ward, 2002).
Recent studies of permeable marine sediments (Gao et al., 2010) and soils (Bateman and Baggs, 2005; Morley et al., 2008)
also observed significant $N_2$ production in the presence of $O_2$ and attributed it to aerobic denitrification

**4.3 DNRA favored under less reducing conditions**

It is generally assumed that strongly reducing conditions favor DNRA over denitrification, yet in our study,
particularly at elevated $O_2$ concentrations, DNRA rates were higher than those of denitrification. That DNRA often seems to
be more important under true anoxic conditions may therefore not be linked directly to the absence of $O_2$ and differential $O_2$
inhibition levels of the two nitrate-reducing processes. Indirect mechanisms are likely to be important. For instance, $H_2S$
accumulation, which often accompanies prolonged anoxia, can inhibit denitrification and simultaneously enhance DNRA
(An and Gardner, 2002; Rysgaard et al., 1996). Another indirect, redox-dependent factor may be the availability of nitrate.
Higher DNRA rates were observed under more $NO_3^-$-limiting conditions induced by prolonged anoxia, probably because
nitrate ammonifiers are able to gain more energy per $NO_3^-$ reduced than denitrifiers (Dong et al., 2011). As nitrate
concentrations are generally much lower under oxygen-free conditions, it appears plausible that anoxia-associated nitrate and
nitrite depletion is conducive to higher DNRA/denitrification rates. While these examples seem to support that DNRA is
favored under true anoxic conditions, results of other studies are more consistent with our observation of higher DNRA than
denitrification rates at elevated $O_2$ concentrations. For example, in estuarine sediments, DNRA was stimulated relative to
denitrification under more oxidizing conditions (Roberts et al., 2014, 2012). The authors argued that DNRA is enhanced by
increasing $Fe^{2+}$ availability at the oxic-anoxic sediment layer during more oxygenated conditions. These studies highlight the
importance of redox conditions in regulating the balance between dentrification and DNRA, however, to what extent $O_2$
directly controls the partitioning between the two nitrate-reducing processes at the enzyme levels remains, to our knowledge,
still unknown. Apparent contradictions with regards to how changing $O_2$ levels may impact nitrate reduction may simply be
due to the counteracting and variable influence of direct versus indirect effects of the variable $O_2$ concentrations.

We cannot fully exclude that through $O_2$ manipulation in this study, we partly affected nitrate-reduction indirectly
through its control of $H_2S$ or $Fe^{2+}$. Yet, we set up the experiments in a way that indirect effects should be minimized (e.g., no
free sulfide in any of the incubations, same organic matter content, same excess $NO_3^-$ concentrations). Hence, this study can
be considered the first experimental investigation into the direct $O_2$ effect on the partitioning between N-loss by
denitrification and N-recycling by DNRA in aquatic sediments. The fact that in our experiments we can essentially exclude





the effects of redox-related parameter changes (i.e., $H_2S$, $NO_3^-$, and $Fe^{2+}$), leads us to the conclusion that in the studied sediments from Lake Lugano, $O_2$ controls the balance between denitrification and DNRA at the organism-level, and that denitrification is in fact more sensitive towards increasing $O_2$ concentrations than DNRA.

**4.4 Direct $O_2$ control on benthic $NO_3^-$ reduction**

It has been previously reported that $O_2$ can either suppress the synthesis of enzymes involved (Baumann et al., 1996) or the enzyme activity itself (Dalsgaard et al., 2014). The observed DIN concentration trends (i.e. decreasing nitrate consumption) with increasing $O_2$ concentrations suggest that the overall activity is modulated mainly at the nitrate reduction step. The differential response of denitrifiers and nitrate ammonifiers further suggests a distinct $O_2$ sensitivity of the nitrate reductase enzymes involved. Denitrifiers and nitrate ammonifiers utilize the same nitrate reductase enzymes (Nar, Nap), and while a differential $O_2$ sensitivity of the same type of enzyme is difficult to explain, it is certainly possible for different enzymes. Indeed, the membrane-bound (Nar) and the periplasmic (Nap) nitrate reductases have distinct affinities towards $NO_3^-$ and $O_2$ tolerance (Mohan and Cole, 2007). Periplasmic nitrate reduction is almost exclusively found in the Proteobacteria and many of the organisms possess both Nar and Nap systems, whose production is regulated in response to ambient $NO_3^-$ and $O_2$ concentrations (Simon and Klotz, 2013). When $NO_3^-$ is scarce, Nap provides a high-affinity (for $NO_3^-$) but low-activity pathway that does not require $NO_3^-$ transport into the cell cytoplasm (Mohan and Cole, 2007). In presence of oxygen, nitrate transport across the cell membrane is repressed, preventing nitrate reduction by the membrane-bound enzyme Nar with its cytoplasm-facing active site (Moir and Wood, 2001). In contrast, nitrate reduction in the periplasm is less $O_2$ sensitive, so that microbes possessing and relying on Nap are likely to have an ecological advantage in environments that are subject to $O_2$ fluctuation (Carter et al., 1995). In nature, nitrate reduction by denitrifiers is assumed to be catalyzed primarily by Nar (Richardson et al., 2007), while most nitrate ammonifiers seem to use Nap (Mohan and Cole, 2007).

Clearly, more fundamental research is required in environmentally relevant non-model microorganisms or mixed communities, to understand better the combined effects of $O_2$ on the nitrogen-transforming metabolic pathways and their regulation. Additional $O_2$ inhibitory effects at one of the down-cascade enzyme levels (Nir, Nrf, Nor, Nos) are likely to exhibit variable $O_2$ sensitivities (Baumann et al., 1996, 1997; Körner H. and Zumft, 1989; Poock et al., 2002). Our observations of higher DNRA/denitrification ratios at higher $O_2$, however, provides at least putative evidence that microorganisms performing DNRA using Nap may be more $O_2$-tolerant than denitrifiers using Nar.

**4.5 Implication for N-elimination versus N-recycling in lakes with fluctuating $O_2$ conditions**

The redox-sensitive partitioning of nitrate elimination (through $N_2$ production by denitrification) versus fixed-N recycling (by nitrate ammonification) has likely important ecosystem-scale consequences. The annual water column turnover of holomictic lake basins such as the south basin of Lake Lugano plays an important role in regulating the contribution of N-removal and N-recycling in the water column (Lehmann et al. 2004; Wenk et al, 2014). To which extent $O_2$ fluctuations



affect N transformation reactions within the sediments remains uncertain. Winter water column turnover ventilates the
bottom waters and re-oxygenates surface sediments that were anoxic for several months. Hence, at least in the top
millimeters of the sediment column we can expect changes in the benthic N cycling. Based on our incubation experiments,
the $O_2$ inhibition threshold was lower for denitrification than for DNRA, possibly reflecting differential adaption of the in
situ microbial community of denitrifiers and nitrate ammonifiers to fluctuating $O_2$ conditions of bottom waters. Indeed, many
nitrate ammonifiers possess both nitrate reductase enzymes (Nap and Nar) and can switch between the two respiratory
systems providing them with an ecological advantage over denitrifiers when substrates become limiting (i.e., with regards to
the primary reductant used in energy metabolism; Mohan and Cole, 2007). During oxygenated bottom-water conditions,
within the benthic redox transition zone, nitrate-reducing microbes at the sediment-water interface will be exposed to
elevated $O_2$ concentrations, similar to the ones tested here. Our experimental data imply that then, at least in the uppermost
sediments, DNRA is favored over denitrification. We may even expect an $O_2$-regulated zonation of DNRA and
denitrification. As a consequence, when denitrification-driven nitrate-reduction is pushed down, it is possible that $NO_3^-$ will
be partially consumed through DNRA before it gets to the "denitrification layer", as nitrate ammonifiers are less $O_2$ sensitive
than denitrifiers. In contrast, denitrification is likely to be a more important nitrate-reducing process compared to DNRA
during water column stratification (suboxia/anoxia of bottom waters), when the sediments are fully anoxic.

365            In the discussion thus far, we implicitly assume that the main control $O_2$ exerts on the absolute and relative rates of
denitrification and DNRA is due to its inhibitory effects at the organism-level, yet the effect of $O_2$ on the coupling of
nitrification and nitrate reduction by either denitrification or DNRA remained unaddressed. Oxygen fluctuations in the
natural environment will affect nitrate regeneration by nitrification, and hence determine how much nitrate is available for
microbial reduction. It has been shown previously that through oxygenation events (e.g., the increase in bottom water $O_2$
concentrations during episodic mixing/ventilation), the overall benthic N elimination in lakes may be enhanced through
coupled nitrification-denitrification, at least transiently (Hietanen and Lukkari, 2007; Lehmann et al., 2015). So, while the
direct effect of elevated $O_2$ would be to hamper fixed N elimination by denitrification at the organism-level, the oxygenation
of previously ammonium-laden but nitrate free (pore-) waters would help to better exploit the benthic nitrate-reduction
potential by increasing the nitrate availability for nitrate-reducing microbes within the sediments, so that the overall nitrate
reduction may be stimulated (Lehmann et al. 2015). Yet, as shown in the present study, oxygenation of the water column and
the upper surface sediments may also act to shift the balance between denitrification and DNRA towards DNRA, thus
promoting N-recycling rather than fixed-N elimination through denitrification. Total nitrification rates were not measured in
this study, but nitrate concentration changes in sediment slurries suggest that at elevated $O_2$ levels there is at least some
production of nitrate. There is no obvious reason to assume that $O_2$-stimulation of the coupling of nitrification and
denitrification on the one hand, and of nitrification and DNRA on the other would per se be different. Yet, as demonstrated
here, DNRA appears to be less $O_2$ sensitive compared to denitrification. It is thus reasonable to expect a higher coupling of
nitrification with DNRA than with denitrification during oxygenated bottom-water conditions. Indeed, there is putative





evidence for such an indirect link between $O_2$ and elevated coupled nitrification-DNRA. In a recent study with estuarine sediments, stronger stimulation of DNRA compared to denitrification was observed during oxygenation of bottom waters, in

parts attributed to the coupling to nitrification (Roberts et al., 2012). Additional experimental work is required to better understand the role of nitrification in regulating the balance between benthic denitrification and DNRA during oxygenation of bottom waters.

It is important to understand that in the natural environment, $O_2$ will not be the only regulator of the balance

between denitrification and DNRA. As previously mentioned, the partitioning of the two nitrate-reducing processes can also be modulated by the substrate (e.g., $NO_3^-$, $NO_2^-$, TOC, $H_2S$, $Fe^{2+}$) availability. The latter may be redox controlled or not. Such regulation may be linked to the differential substrate affinity of the two processes when competing for the same electron acceptor (e.g., nitrate/nitrite) providing selective pressure that can drive communities either towards denitrification or DNRA (Kraft et al. 2014), or simply due to differing substrate requirements in the case of chemolithotrophic versus

organotrophic nitrate reduction.

For example, nitrate concentrations in the water column of the lake sampled in this study (Lake Lugano) varied significantly over the year, with very low $NO_3^-$ concentrations during the stagnation period (during anoxia) (Fig. 1). As a consequence, it is reasonable to assume that the relative partitioning between denitrification and DNRA in a natural

environment is affected by the fluctuating nitrate concentrations (e.g., Tiedje et al., 1988, Dong et al., 2011). Similarly, $Fe^{2+}$ levels in near-bottom waters and sediment porewaters in Lake Lugano are greater during the anoxia/stratification period (Lazzaretti et al., 1992). At least in environments where chemolithotrophic processes contribute to the overall nitrate reduction, such redox-dependent $Fe^{2+}$ concentration changes (or changes of other electron donors such as $HS^-$) may affect the balance between DNRA and denitrification (e.g., Robertson et al. 2015). Hence, in addition to the direct regulating effects of

$O_2$ on the partitioning between denitrification and DNRA, which we have demonstrated here experimentally, $O_2$ can act as indirect regulator of fixed N elimination versus regeneration. The ultimate ecosystem-scale DNRA/denitrification ratio in environments that are subject to $O_2$ fluctuating conditions is difficult to predict, because direct and indirect $O_2$ regulation may act concomitantly and in opposite ways.

## 5 Conclusion

The presented results broaden the range of $O_2$ inhibition thresholds of benthic denitrification at micromolar $O_2$ levels, demonstrating that benthic denitrification may resist full inhibition up to almost 80 µM $O_2$. Similarly, sedimentary DNRA does not necessarily require true anoxia, and was even less sensitive than denitrification to higher $O_2$ levels. Our data suggest that the balance between DNRA and denitrification is modulated by $O_2$ at the nitrate-reducing enzyme level (Nar versus Nap). However, more in-depth investigations on the exact role of oxygen in regulating other denitrification and/or





nitrate-ammonification enzymes in microbial pure culture experiments are needed. The differential tolerance of denitrifiers versus nitrate ammonifiers towards $O_2$ has important implications for natural environments with fluctuating $O_2$ conditions. Based on our results, one might argue that DNRA may be more important during phases of bottom-water oxygenation, while anoxic conditions during the stratification period may favor full denitrification to dinitrogen. Whether and when fixed nitrogen is preserved in a lake or eliminated by denitrification is, however, difficult to predict, as this will depend also on multiple indirect effects of changing $O_2$ levels. For example, nitrification and the redox-dependent modulation of substrates that may be relevant for denitrification or DNRA (such as nitrite, the substrate at the branching point between the two processes, and/or sulfide as potential inhibitor of denitrification and stimulator of chemolithotrophic DNRA) will play an important role both with regards to the overall nitrate reduction rate, as well as the balance between different nitrate reducing processes. Internal eutrophication from N in high-productivity lakes is generally less of a concern than from P. Nevertheless, it needs to be considered that oxygenation may reduce the overall fixed N-elimination capacity of the bottom sediments by impairing denitrification more than DNRA, partially counteracting the generally positive effects of hypolimnetic ventilation in the context of benthic nutrient retention/elimination, and with implications on the nutrient status in the water column.

**Author contribution**

JZ and MFL initiated the project. ANYC performed all sample collection and conducted the experimental work with help from AG. FL provided the water column chemistry profiles. ANYC, JZ and MFL performed data analysis and interpretation. ANYC and MFL prepared the manuscript with input from all co-authors.

**Competing interest**

The authors declare that they have no conflict of interest.

**Acknowledgements**

We thank Thomas Kuhn for technical support in the laboratory and Stefano Beatrizotti, Maciej Bartosiewicz, Guangyi Su and Jana Tischer for assistance during sampling on the lake. We also thank Elizabeth Robertson and Bo Thamdrup for their help during the development of the method for slurry incubation experiments. The study was funded by the Swiss National Science Foundation (SNF) project 153055, granted to Jakob Zopfi and Moritz F. Lehmann. We are also grateful to the Freiwillige Akademische Gesellschaft (FAG) Basel that also financially supported the study.

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





**Table 1: Overview of N transformation rates in O$_2$-controlled slurry incubation experiments. Negative and positive values correspond to net NO$_3^-$ or NH$_4^+$ consumption and production rates over incubation time, respectively. Standard errors are indicated in bracket for *n* replicates. Average denitrification and DNRA rates (μmol N g$^{-1}$ wet sed. d$^{-1}$) in anoxic control experiments were: 0.11 ± 0.01 and 0.12 ± 0.04, respectively, at Figino; 0.12 ± 0.01 and 0.11 ± 0.01, respectively, at Melide.**

| Sampling site | O$_2$ conc. in slurry μmol L$^{-1}$ | *n* | Inhibition compared to control (%) Denitrification | DNRA | NO$_3^-$ | NH$_4^+$ |
|---|---|---|---|---|---|---|
| | | | | | μmol N g$^{-1}$ wet sed. d$^{-1}$ | |
| Figino | 0 (control) | 12 | | | -1.4 (0.1) | 0.7 (0.03) |
| | 1.2 | 3 | 29 | 51 | -0.8 (0.2) | 0.8 (0.4) |
| | 2 | 3 | 57 | 35 | -0.4 (0.1) | 0.8 (0.2) |
| | 2.8 | 3 | 68 | 17 | -0.4 (0.3) | 0.7 (0.1) |
| | 3.4 | 2 | 64 | 29 | -0.6 (0.2) | 0.6 (0.1) |
| | 4.1 | 3 | 77 | 45 | -0.9 (1.3) | 0.5 (0.2) |
| | 8.6 | 3 | 85 | 60 | -1.1 (0.3) | 0.4 (0.0) |
| | 16 | 4 | 70 | 84 | -0.2 (0.5) | 0.1 (0.2) |
| | 24.1 | 3 | 86 | 77 | -0.2 (0.6) | 0.2 (0.1) |
| | 38 | 3 | 93 | 39 | 0.2 (2.1) | 0.0 (0.2) |
| | 61.3 | 3 | 94 | 64 | -0.3 (0.3) | -0.2 (0.1) |
| | 78.6 | 6 | 96 | 79 | 1.1 (1.4) | -0.1 (0.0) |
| | | | | | | |
| Melide | 0 (control) | 9 | | | -1.0 (0.4) | 0.2 (0.0) |
| | 0.8 | 2 | 43 | 37 | -0.7 (0.2) | -0.1 (0.1) |
| | 1.8 | 2 | 63 | 53 | -0.6 (0.1) | 0.2 (0.2) |
| | 2.9 | 3 | 61 | 58 | -0.5 (0.3) | -0.1 (0.2) |
| | 3.8 | 4 | 58 | 65 | -0.2 (0.1) | -0.1 (0.3) |
| | 4.9 | 3 | 74 | 64 | -0.3 (0.2) | -0.1 (0.3) |
| | 9 | 7 | 73 | 69 | 0.0 (0.1) | -0.1 (0.2) |
| | 13.1 | 2 | 69 | 37 | -0.6 (0.1) | -0.1 (0.0) |
| | 21.3 | 2 | 66 | 56 | -0.4 (0.1) | -0.1 (0.1) |
| | 44.4 | 2 | 67 | 34 | -0.3 (0.2) | -0.1 (0.2) |
| | 58.6 | 3 | 91 | 60 | -0.1 (0.1) | -0.1 (0.2) |
| | 73.2 | 4 | 93 | 75 | 0.2 (0.2) | -0.4 (0.1) |


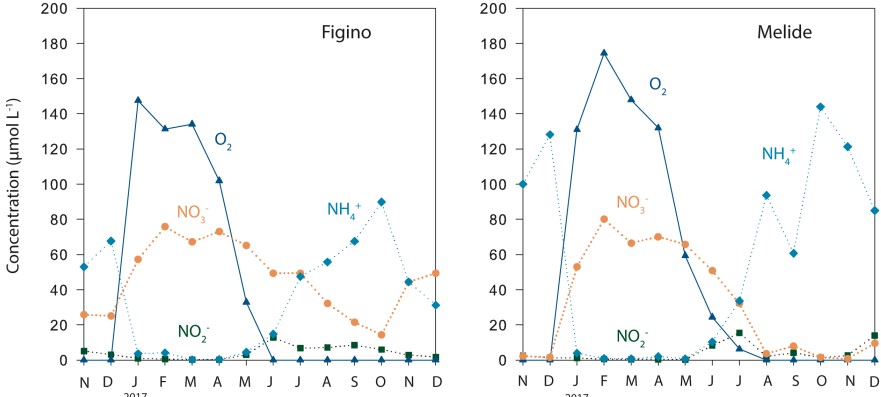

**Figure 1: Concentrations of dissolved O$_2$ and reactive nitrogen in the bottom waters (2 m above the sediments) of the Lake Lugano**
**South Basin in 2017.**

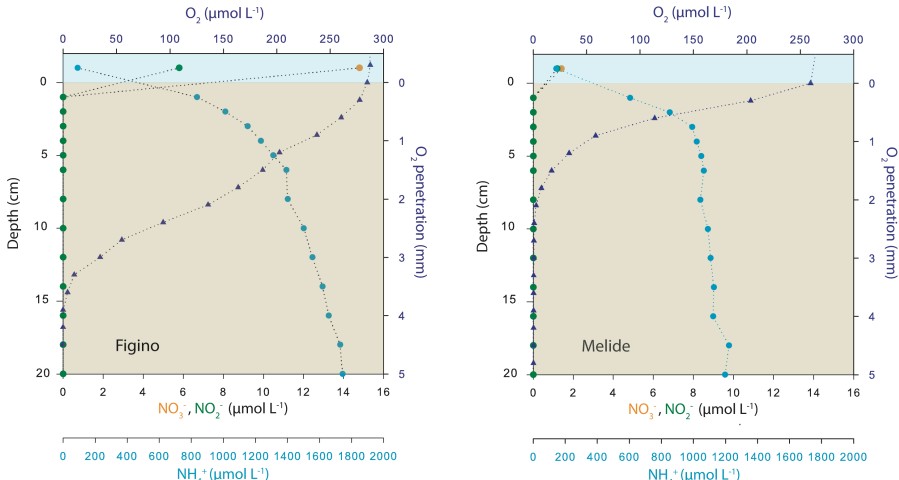

**Figure 2: Ex-situ sediment porewater profiles (O$_2$ and dissolved inorganic nitrogen) at the two sampling stations of the Lake Lugano South Basin in a sediment core collected in October 2017. Oxygen concentration profiles (note different depth units) were determined in aerated cores, and thus are representative of the O$_2$ penetration during aerated conditions in the water column, as**
**seen between January and April (see Fig. 1).**





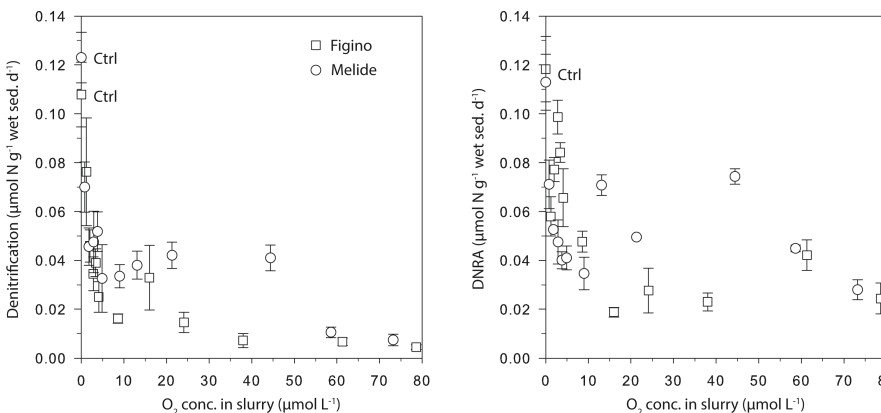

**Figure 3: Denitrification and DNRA rates as a function of dissolved $O_2$ concentration in dilute sediment slurry from Figino and Melide. Error bars represent the standard error of n replicate experiments and measurements (Table 1).**


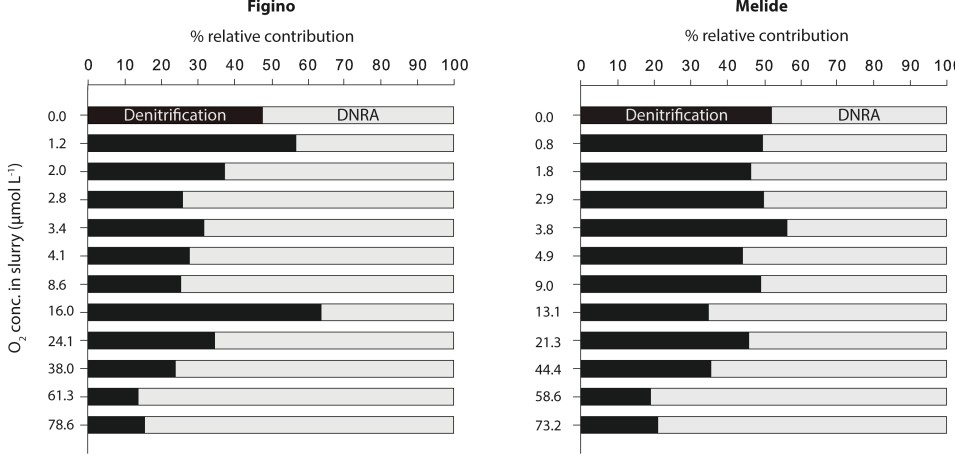

**Figure 4: Relative contribution (%) of denitrification and DNRA to total nitrate reduction under variable $O_2$ conditions.**
