# Peer review of "Direct O2 control on the partitioning between denitrification and dissimilatory nitrate reduction to ammonium in lake sediments"

_Biogeosciences, 2019_

## Referee Comment (RC1) · Anonymous Referee #1 · 13 Aug 2019

The article by Cojean et al deals with the effect of oxygen concentrations on the partitioning between denitrification and dissimilatory nitrate reduction to ammonium in a lake sediment. The study uses stable isotopes in anaerobic slurries amended with increasing concentrations of oxygen to measure potential rates of the different processes.

I have several methodological and interpretation questions. Also I think the data could be used to obtain more information in the processes studied. many of the questions left open are supposed to be in another article in prep (Cojean in prep). Maybe the authors should consider to include these data here.

[Figure]

More details comments can be found below.

L74. "aquatic", you mean limnetic?

L89. "ventilated", waters are oxygenated but not ventilated

L97-99. How do authors explain the presence of nitrate under anoxic conditions?

L102. At which temperature was the profiling made?

L104. How were the cores sectioned ? Under N2 or normal atmosphere? Please add information on time, temperature and xg of centrifugation.

L109. I understand that all manipulations were performed under normal atmosphere.

L111. Can you provide a reference for the artificial lake water?

L115. Flushing and preincubating overnight have an impact on the presence of other gases (N2O, CO2 etc) as well as on the availability of labile organic matter, both dissolved and particulate. A comment on these limitations of sediment slurries should be added in my opinion.

L117. No information on how much nitrate was added is given. How does it compare to the natural concentrations? If the concentrations used are not saturating for the enzymes in question, relative rates can be dependent on the enzymatic kinetics of each process.

L118. Details should be added on how the duration of the experiments and time points were determined. Was it based on the oxygen evolution? On some previous experiments? How did the authors know in advance that the incubation should last approx 10 h?

L121. Any particular reason for the difference in the compensation method? If transfer of liquids was performed under normal atmosphere how did the authors account for any oxygen contamination? How were samples poisoned to prevent microbial activity?

L125. Monitoring of O2 concentrations was possible with the optode spots. It is important to know the degree of the "marked decline" was observed and have an estimate of the possible effect on the rates (by looking at the corresponding N species concentrations).

L132. Gordon reference is missing.

L140. Did the authors measure organic carbon content which is an important factor in the partitioning between DN and DNRA?

L147. Rates were determined based on initial and final data? or regressions over time?

L162. Maybe I missed it somewhere but I did not find any statistical analysis of the data.

L169. The profiles do indeed indicate that the surface sediment of the Figino is less active at the surface than further below. Any reason why? However, the profiles of nitrate do not indicate directly that the sediment acts as a sink given that nitrate and nitrite microprofiles occur same as for oxygen at a mm scale.

L180. A treatment with ATU to block nitrification would have clarified many of the issues mentioned here.

L185 & table 1. Rates of 1.1 umol N g d are very low and probably close to the detection limit of the methods used. Data on detection limits should be added in M&M. This is critical when standard error of 6 replicates is 130% of the mean value.

L192. The authors could take advantage of their data and provide kinetic rate data calculations to allow for easier comparisons with other studies. Also given the big dispersion of the data it would also provide more quantitative information on the differences between sites and processes. Error of percentages should also be provided L196.

**BGD**

L214. reduced should be consumed. There is a strong discrepancy as the authors mention between the budget and the order of N species transformations based on stable isotopes and total rates.

L217 authors claim that biotic immobilization can be an explanation, however, previously they mention that N species exchangeable fraction was insignificant.

L224. Authors should try to put into quantitative perspective their findings given that they have oxygen concentration data for bottom water.

L237. The effect of oxygen concentration on the partitioning between nitrate reduction will only have an impact on a um scale.

L280. Author should show the time evolution of oxygen in the slurries to prove no microniches were created in the slurries. Just mentioning that it is unlikely they were formed is speculation. Where was the spot placed? How many per bottle? Was it always under water? Was any sort of beads were included in the bottles?

L285. This makes no sense as pelagic communities are more easily exposed to high oxygen concentrations and thus should be more tolerant, rather than the opposite.

L292. These sections (4.3-4.5) are purely based on speculations and are not based on any data from this study. Actually these sections just highlight the limitations of the study.

L314. How did the authors ensure lack of H2S in the slurries forming? is is by eliminating SO4 presence or its formation somehow?

L320. Authors have no data to discuss the partitioning between the activity of NAP and NAR enzymes.

L349. Authors could try to quantify the effect. Is it significant?

L412. The data do not differentiate in any way between NAR and NAP.

Fig 2. Are data single measurements or mean of several replicates? Oxygen in water column of panel b is missing. Y axis scale is not "oxygen penetration depth" but just depth, same as in the first y axis. In the fully mixed lake conditions, oxygen concentrations does not increase above 150 uM yet in the microprofiles O2 is 250 uM.

Fig 3. I thin it is better to show the rate data of the same site in the same panel as in Fig 4. Also use different colours to be able to differentiate between thte two.

Fig 4. Mean values can be represented as points with their appropriate error bars plus a statistical analysis to confirm significant differences.

---

## Referee Comment (RC2) · Anonymous Referee #2 · 14 Aug 2019

Cojean and Co workers investigate the partition between DNRA and denitrification at different oxygen concentrations in slurries prepared from lake sediments. From their experimental data they conclude that that No3 reduction rates (here DNRA and denitrification rates) are generally reduced in the presence of oxygen as compared to rates measured at anoxic conditions, but that nitrate reduction is still going on at oxygen concentrations ranging from 0.8 to 78.6 $\mu$M. The authors further conclude from their experimental data that the partition of nitrate reduction between denitrification and DNRA differs at different oxygen concentrations, so that the relative importance of DNRA increases with increasing oxygen concentrations. In my classical (and perhaps narrow minded?) view the conclusion on aerobic nitrate reduction and on the partition between

DNRA and denitrification in relation to oxygen is still a bit controversial. Therefore I would like the authors to present more information about the actual conditions of their experiments, and apply stronger statistics, as well as present the data from their statistical analyses (it is mentioned that t-tests were used, but no results (p values etc) are presented.

1. The experiments were performed as slurry incubations in serum bottles to which 15no3 were added. What was the resulting 15NO3 concentration in the slurries? 2. Oxygen was added to the headspace in the slurries and monitored with Oxygen Sensor Spots during the course of the experiments. Please present the data on oxygen concentrations in the serum bottles, trough out the course of the experiment. Is the oxygen concentration constant during the course of the experiment for all treatments or does the concentrations drops to critical levels in some of those? Present those data eventually in the supplementary information. 3. The rates of DNRA and denitrification were calculated from the accumulation of 15N2 and 15NH4 over the course of an incubation period of 10 hours. Please present data that shows how the concentration of the isotopes changed during the course of the experiment. Do you see a linear increase in the concentration of the isotopes as function of time during the entire incubation period? Can you document such an increase eventually through linear regression analysis? Point 2 and 3 are mandatory for a reliable interpretation of the data. The optimal situation is a) The oxygen concentration does not drop significantly (or not critically ) during the course of the experiment. b) There is a linear increase in the concentration of the isotopes as function of time. An eventual derivation from this situation can compromise the validity of the experiment and the conclusions that can be drawn from the data.

4. Statistics is a strong tool for getting sound scientific statements. It is mentioned that t-tests were used to test for differences DNRA and denitrification rates, but no data from these tests are shown. Pleases report those statistical data (eventually in table 1). I would also recommend the use of a statistical method that investigate if the partition of DNRA and denitrification, differs significantly at different oxygen concentrations. Use eventually an ANOVA analysis. You might e.g. use DNRA/(DNRA+Denitrification) i.e. the proportion of DNRA to the measured nitrate reduction rate, as test variable. I understand that the experiment was performed, with replicates and that both rates of denitrification and DNRA were measured simultaneously in the same serum bottle. So it should be possible do a sound statistical analysis. Such an analysis can only strength your interpretation and conclusions. Alternatively use a correlation analysis, where you investigate for significant (positive/negative) correlation, between e.g. DNRA/(DNRA+Denitrification) and the oxygen concentration.

5. A comment to the statement l.185. It is stated that the high background of 14NH4 prevent the 15NH4 from becoming nitrified, and that the isotope derived rate of DNRA, therefore is not underestimated due to nitrification. I do not think that this argument is valid. The problem is the same as for other tracer studies like S35 based studies of sulfate reduction or 14C based studies of e.g. methane turnover, where you have production and consumption occurring simultaneously. Moeslund et al. (1994) showed with experimental data that Sulfate reduction rates as measured with radiotracers, added to the experimental system at very low concentrations was underestimated if sulfide oxidation was present. Xiao et al. (2018) showed from modeling of a tracer study that the degree of underestimation of rates of methane production was proportional to the incubation period in systems with methane production and methane oxidation. I sugget therefore that you delete this statement. Note that if your overall conclusions regarding DNRA and denitrification and oxygen are correct, an eventual underestimation of DNRA at high oxygen concentrations, would not compromise that conclusion. Moeslund, L., B. Thamdrup, and B. Barker Jørgensen. 1994. Sulfur and iron cycling in a coastal sediment: Radiotracer studies and seasonal dynamics. Biogeochemistry 27: 129-152. Xiao, K.-Q., F. Beulig, H. Røy, B. B. Jørgensen, and N. Risgaard-Petersen. 2018. Methylotrophic methanogenesis fuels cryptic methane cycling in marine surface sediment. Limnol. Oceanogr. 63: 1519-1527.

---

## Author Comment (AC1) · 30 Aug 2019

Response to comments by Anonymous Referee #1

I have several methodological and interpretation questions. Also I think the data could be used to obtain more information in the processes studied. many of the questions left open are supposed to be in another article in prep (Cojean in prep). Maybe the authors should consider to include this data here.

Reply: We thank the reviewer for her/his valuable inputs. We will make changes in the revised manuscript accordingly. In particular, we will consider mentioning some of

the results from the companion paper, which we plan to submit soon. In any case, the scope of the two articles is very different, and the main focus of the other paper is on the role of inorganic electron donors (e.g. Fe2+, H2S) in regulating the partitioning between denitrification and DNRA. As Fe2+ and H2S levels, as well as the presence of NOx itself, are redox-dependent, O2 may indirectly affect denitrification and DNRA rates simply by affecting the Fe2+, H2S or NOx concentrations. We will stress this point in the revised manuscript, but we prefer not to discuss these aspects in greater detail, as they will be part of the paper on Fe2+/H2S-modulation.

L74. "aquatic", you mean limnetic?

Reply: No, we actually meant aquatic in general. To our knowledge, there are no experimental studies on the systematic O2 control on DNRA (e.g. continuous control of the O2 concentration over time) with samples from an aquatic environment (e.g. marine or freshwater). Most of the time, in previous research, measured DNRA rates were simply correlated with the in situ O2 concentrations.

L89. "ventilated", waters are oxygenated but not ventilated

Reply: We will change that.

L97-99. How do authors explain the presence of nitrate under anoxic conditions?

Reply: Nitrate diffuses down to bottom waters, where it is consumed by denitrification after the onset of denitrification in June. Thereafter, concentrations decrease, but nitrate is not used up completely, simply because the duration of anoxia until spring mixing is not long enough. In the ocean, the presence of nitrate in oxygen-free water column zone is the norm rather than the exception.

L102. At which temperature was the profiling made?

Reply: At room temperature ($\sim20°C$).

L104. How were the cores sectioned ? Under N2 or normal atmosphere? Please add

information on time, temperature and xg of centrifugation.

Reply: Cores were sectioned under normal atmosphere. Sediment sections were centrifuged for 10 min at 4700 rpm (room temperature). This information will be included in the revised manuscript.

L109. I understand that all manipulations were performed under normal atmosphere.

Reply: No, sampling of gas and liquid samples were performed in a glovebox under N2 atmosphere to avoid any O2 contamination. Also, before incubations, the slurries were preincubated overnight to remove any potential traces of O2. Additionally, N2 contamination was avoided by flushing the gas-tight syringe several times with He prior to sampling. The procedures and precautions to avoid O2 or N2 contamination are routine in our labs.

L111. Can you provide a reference for the artificial lake water?

Reply: Yes, this will be included in the revised manuscript. The reference is: Smith E.J., Davison W. and Hamilton-Taylor J. (2002) Methods for preparing synthetic freshwaters. Wat. Res. 36, 1286-1296.

L115. Flushing and preincubating overnight have an impact on the presence of other gases (N2O, CO2 etc) as well as on the availability of labile organic matter, both dissolved and particulate. A comment on these limitations of sediment slurries should be added in my opinion.

Reply: We are not sure whether purging will significantly affect the DOC pool, which consists mostly of longer carbon-chain molecules rather than volatiles. But certainly, the interference with in situ conditions in general is non-negligible. For example, larger aggregates in the sediments might have been disrupted altering microbe-particle interactions. Also, the pH may have been changed slightly. We will mention this aspect in the revised manuscript.

L117. No information on how much nitrate was added is given. How does it compare

to the natural concentrations? If the concentrations used are not saturating for the enzymes in question, relative rates can be dependent on the enzymatic kinetics of each process.

Reply: This information will be included in the revised manuscript. The final NO3- conc. in slurry was $\sim$ 120 $\mu$mol L-1. The experimental concentration was thus higher than the in situ one and significantly higher than the half saturation concentrations for either of the nitrate-reducing processes.

L118. Details should be added on how the duration of the experiments and time points were determined. Was it based on the oxygen evolution? On some previous experiments? How did the authors know in advance that the incubation should last approx 10 h?

Reply: This information will be added in the revised manuscript. Preliminary tests were performed in order to assess the minimal incubation time required to obtain a measurable signal for 15N-N2 measurement through mass spectrometry. Initially, the incubation lasted for 24 h, however, the precise monitoring of O2 conc. over such a long time was quite difficult and NO3- conc. dropped quite rapidly, which may have affected the partitioning between the two processes. Therefore, we tested several incubation periods (15 h, 10 h), and ultimately decided to go with 10 h incubations. This way, we were able obtain clear and robust 15N-N2 signals and control of O2 conc. in slurries in parallel experiments was easier to manage.

L121. Any particular reason for the difference in the compensation method? If transfer of liquids was performed under normal atmosphere how did the authors account for any oxygen contamination? How were samples poisoned to prevent microbial activity?

Reply: We decided to use 2 different compensation methods for T0 and other time points in order to avoid, as much as possible, the dilution of the residual nutrient and gas pools by milli-Q and He addition, respectively. At T0, He was used because we collected a greater liquid volume (6 mL) in order to measure nutrient concentrations and

DNRA rates, while at T1 and T2, only a 2 mL gas sample was taken in exchange with anoxic milli-Q instead of He. At T3, the incubation was stopped and we did not need to compensate for pressure changes inside the vials anymore, therefore no liquid/gas was added. As mentioned above, all liquid and gas samples were performed in a glovebox under N2 atmosphere to prevent O2 contamination, and additional contamination of N2 was prevented by flushing the syringe several times with He prior to sampling. Samples were not poisoned. Liquid samples were filtered (0.2 $\mu$m) and stored at -20°C.

L125. Monitoring of O2 concentrations was possible with the optode spots. It is important to know the degree of the "marked decline" was observed and have an estimate of the possible effect on the rates (by looking at the corresponding N species concentrations).

Reply: A table showing the exact O2 concentration measured in slurry at different time points during the incubation will be included in the supplementary information. Oxygen concentration did never drop below 85% of the initial targeted O2 concentration. We will therefore replace "marked decline" by "decline" in the revised manuscript.

L132. Gordon reference is missing.

Reply: Thanks, it will be added.

L140. Did the authors measure organic carbon content which is an important factor in the partitioning between DN and DNRA?

Reply: We did not measure the organic carbon (OC) content during these incubation experiments, but we do not think that changes in OC content over the incubation time played an important role in regulating the partitioning between the two processes in this particular case. In general, the most important factor to consider is the ratio OC/ NO3-, where DNRA is favored at high OC/ NO3- (e.g. NO3–limiting) ratios, and denitrification at lower OC/NO3- ratios (e.g OC-limiting; van den Berg et al., 2016). We agree that changes in OC content over the incubation time likely varied among the different

O2 treatments (e.g. higher OM remineralization rates at higher O2 conc.), however, OC concentration measurement in slurry incubation experiments within the frame of another study using sediments from the same setting displayed relatively high OC concentrations (initial OC concentration $\geq$ 510 mg/L; Cojean et al., in prep.), suggesting that OC was never limiting over the 10 h incubation period (8°C). Similarly, NO3- was always present in excess during the entire incubation experiment (maximum decrease in NO3- conc. of about 8 $\mu$mol L-1 from T0 to Tend). Hence, as neither NO3- nor OC were limiting, it is likely that the partitioning between denitrification and DNRA in the different O2 treatments, as compared to control incubations, was not so much affected by changes in the OC/ NO3- but rather by O2. van den Berg E.M., Boleij M., Kuenen J.G., Kleerebezem R. and van Loosdrecht M.C.M. (2016) DNRA and denitrification coexist over a broad range of acetate/N- NO3- ratios in a chemostat enrichment culture.

L147. Rates were determined based on initial and final data? or regression over time?

Reply: Rates were determined through linear regression of 15N-N2 concentrations versus time. We will mention it in the revised manuscript.

L162. Maybe I missed it somewhere but I did not find any statistical analysis of the data.

Reply: We will include results of the statistical analyses in the revised manuscript.

The profiles do indeed indicate that the surface sediment of the Figino is less at the surface than further below. Any reason why? However, the profiles of nitrate do not indicate directly that the sediment acts as a sink given that nitrate and nitrite microprofiles occur same as for oxygen at a mm scale.

Reply: Less what? It is unclear to us what the reviewer refers to. Anyway, the sediments represent a strong sink for NOx. Nitrate is consumed quantitatively within the first mm, and there is not the slightest indication for net production of NOx, e.g. indicated by a local NO3- maximum.

[Figure]

L180. A treatment with ATU to block nitrification would have clarified many of the issues mentioned here.

Reply: We agree that this would have been a good idea. Nevertheless, we believe that our data set is very much meaningful.

L185 & table 1. Rates of 1.1 umol N g d are very low and probably close to the detection limit of the methods used. Data on detection limits should be added in M&M. This is critical when standard error of 6 replicates is 130% of the mean value.

Reply: The detection of the method is on the order of 0.02 $\mu$mol L-1. We will mention this in the M&M section.

L192. The authors could take advantage of their data and provide kinetic rate data calculations to allow for easier comparisons with other studies. Also given the big dispersion of the data it would also provide more quantitative information on the differences between sites and processes. Error of percentages should also be provided L196.

Reply: Given the interference with in situ condition, and the addition of substrates in excess, we do not think that the absolute rates, which must be considered potential rates, matter in the context of this study. We specifically look at the relative rates, relative to controls, and relative with regards to the ratio of denitrification and DNRA. As mentioned above, we will provide information on the error of the percentage numbers.

L214. reduced should be consumed. There is a strong discrepancy as the authors mention between the budget and the order of N species transformations based on stable isotopes and total rates.

Reply: We will replace "reduced" with "consumed".

L217 authors claim that biotic immobilization can be an explanation, however, previously they mention that N species exchangeable fraction was insignificant.

Reply: We referred to the abiotic exchangeable fraction through NH4+ adsorption to the sediment, which we consider negligible. Yet, we cannot exclude completely the biotic immobilization of NO3- in NO3–storing microorganisms, especially since we observed the presence of Beggiatoa sp., which are able to store significant amounts of NO3- in the studied sediments (Cojean et al., in prep.).

L224. Authors should try to put into quantitative perspective their findings given that they have oxygen concentration data for bottom water.

Reply: It is tempting to extrapolate our experimental results to the situation in the water column, and thus assess, for example, expected changes in the denitrification/DNRA ratio with changing O2 concentrations in the near-bottom waters. While there will be a more or less direct link between the bottom water O2 concentrations and the O2 penetration within sediments, we can only speculate about what the O2 concentrations will be right at and below the sediment-water-interface over the annual cycle. In any case, independent of the O2 in bottom waters, O2 will be consumed to completion by aerobic respiration and the oxidation of reduced substances within the first mm of the sediment column.

L237. The effect of oxygen concentration on the partitioning between nitrate reduction will only have an impact on a um scale.

Reply: See last comment. We agree that O2 will only regulate the partitioning between the two nitrate reduction processes in the surficial sediments. Upscaling it to the entire lake-basin is problematic.

L280. Author should show the time evolution of oxygen in the slurries to prove no microniches were created in the slurries. Just mentioning that it is unlikely they were formed is speculation. Where was the spot placed? How many per bottle? Was it always under water? Was any sort of beads were included in the bottles?

Reply: We will upload a table with the O2 conc. measured over time in the supplementary information. Oxygen spots were placed on the glass wall of each serum bottle and the spot remained under water during measurement. No beads were included, but we shook the bottles manually every 30 minutes, prior to O2 measurement, and between O2 readings, the bottles were placed on a shaker (80 rpm). As mentioned above, we did not observe any dramatic decline of O2 conc. in any of the experiments and as the slurries were strongly diluted, we think that microniches did not form.

L285. This makes no sense as pelagic communities are more easily exposed to high oxygen concentrations and thus should be more tolerant, rather than the opposite.

Reply: Good point, and we agree that this is speculation. We will delete the statement.

L292. These sections (4.3-4.5) are purely based on speculations and are not based on any data from this study. Actually, these sections just highlight the limitations of the study.

Reply: We do not agree. In 4.3, for example, we provide conclusive experimental data that indicate that, in relative terms, DNRA is favored over denitrification under less reducing conditions. Going one step further, true, we speculate about the reasons for our observational data, comparing our experimental data with observations from other natural environments. This way we test plausibilities. These sections are part of an honest scientific discussion, where we try to explain the observations we made. Yes, there is a certain degree of speculation involved, and we cannot always provide final answers. Fair enough. But speculative parts are clearly highlighted as such, and it seems appropriate to discuss the potential implications of our results, even if we cannot provide conclusive evidence for all the statements. We agree that our study has limitations, and we do not hide them. And these limitations may become most evident in these speculative sections the reviewer refers to. But we opt to leave them in, maybe rephrased even more cautiously, as they are intended to stimulate future work that will verify/falsify the putative regulating mechanisms proposed.

L314. How did the authors ensure lack of H2S in the slurries forming? is is by eliminating SO4 presence or its formation somehow?

Reply: We measured free H2S concentrations and they were below or close to the detection limit. We will mention this point in the revised manuscript.

L320. Authors have no data to discuss the partitioning between the activity of NAP and NAR enzymes.

Reply: See comment above. Yes, this is speculative, and we will rephrase this part more carefully, so that it is clear that we speculate. But, in the attempt to explain the results, we find, there is some room for speculation. We observe clear experimental evidence for direct O2 control of denitrification versus DNRA. The most obvious "corner" to look for plausible explanation would be at the enzymatic level, and not mentioning this (even if we cannot provide molecular data) would seem unmindful to us. The differential response of denitrifiers and nitrate ammonifiers at least suggests a distinct O2 sensitivity of the nitrate reductase enzymes involved. Unfortunately, we do not have information on the activity of NAP versus NAR, but in line with what we mentioned above, we would like to propose plausible hypothesis that can then be tested by future work.

L412. The data do not differentiate in any way between NAR and NAP.

Reply: Agreed. Since we do not have any solid data in this regard we will delete "(NAR versus NAP)". But we would like to stick to our more general conclusion that the balance between DNRA and denitrification is modulated by O2 at the nitrate-reducing enzyme level.

Fig 2. Are data single measurements or mean of several replicates? Oxygen in water column of panel b is missing. Y axis scale is not "oxygen penetration depth" but just depth, same as in the first y axis. In the fully mixed lake conditions, oxygen concentrations does not increase above 150 uM yet in the microprofiles O2 is 250 uM.

Reply: The results show the mean downcore concentrations measured in duplicate and triplicate cores for N species and O2 conc., respectively. The second x-axis label

is indeed depth at a different scale, and we will correct it. Concentrations measured in bottom water and overlying water during O2 microprofiling do not match because the microprofiling was not performed under in situ conditions but under aerated conditions (and higher temperature). Thus the O2 profiles reflect the potential biological oxygen demand of the two sampling stations, and the ex-situ O2 penetration depth was meant to serve the biogeochemical characterization of the sediments.

Fig 3. I thin it is better to show the rate data of the same site in the same panel as in Fig 4. Also use different colours to be able to differentiate between thte two.

Reply: We would like to keep the original graph. Indeed, we aimed to show that, independently of the studied site, we observed similar patterns regarding the absolute O2 control on denitrification and DNRA rates. We will, nevertheless, use different colors to differentiate between the two processes.

Fig 4. Mean values can be represented as points with their appropriate error bars plus a statistical analysis to confirm significant differences.

Reply: The statistical results will be included in the manuscript and adequately presented in the figures.

---

## Author Response (AR1)

**Response to comments by Anonymous Referee #1**

I have several methodological and interpretation questions. Also I think the data could be used to obtain more information in the processes studied. many of the questions left open are supposed to be in another article in prep (Cojean in prep). Maybe the authors should consider to include this data here.

We thank the reviewer for her/his valuable inputs. We will make changes in the revised manuscript accordingly. In particular, we will consider mentioning some of the results from the companion paper, which we plan to submit soon. In any case, the scope of the two articles is very different, and the main focus of the other paper is on the role of inorganic electron donors (e.g. $Fe^{2+}$, $H_2S$) in regulating the partitioning between denitrification and DNRA. As $Fe^{2+}$ and $H_2S$ levels, as well as the presence of $NO_x$ itself, are redox-dependent, $O_2$ may indirectly affect denitrification and DNRA rates simply by affecting the $Fe^{2+}$, $H_2S$ or $NO_x$ concentrations. We will stress this point in the revised manuscript, but we prefer not to discuss these aspects in greater detail, as they will be part of the paper on $Fe^{2+}/H_2S$-modulation.

L74. "aquatic", you mean limnetic?

Reply: No, we actually meant aquatic in general. To our knowledge, there are no experimental studies on the systematic $O_2$ control on DNRA (e.g. continuous control of the $O_2$ concentration over time) with samples from an aquatic environment (e.g. marine or freshwater). Most of the time, in previous research, measured DNRA rates were simply correlated with the in situ $O_2$ concentrations.

L89. "ventilated", waters are oxygenated but not ventilated

Reply: We will change that.

L97-99. How do authors explain the presence of nitrate under anoxic conditions?

Reply: Nitrate diffuses down to bottom waters, where it is consumed by denitrification after the onset of denitrification in June. Thereafter, concentrations decrease, but nitrate is not used up completely, simply because the duration of anoxia until spring mixing is not long enough. In the ocean, the presence of nitrate in oxygen-free water column zone is the norm rather than the exception.

L102. At which temperature was the profiling made?

Reply: At room temperature (~20°C).

L104. How were the cores sectioned ? Under N2 or normal atmosphere? Please add information on time, temperature and xg of centrifugation.

Reply: Cores were sectioned under normal atmosphere. Sediment sections were centrifuged for 10 min at 4700 rpm (room temperature). This information will be included in the revised manuscript.

L109. I understand that all manipulations were performed under normal atmosphere.

Reply: No, sampling of gas and liquid samples were performed in a glovebox under $N_2$ atmosphere to avoid any $O_2$ contamination. Also, before incubations, the slurries were preincubated overnight to remove any potential traces of $O_2$. Additionally, $N_2$ contamination was avoided by flushing the gas-tight syringe several times with He prior to sampling. The procedures and precautions to avoid $O_2$ or $N_2$ contamination are routine in our labs.

L111. Can you provide a reference for the artificial lake water?

Reply: Yes, this will be included in the manuscript. The reference is: Smith E.J., Davison W. and Hamilton-Taylor J. (2002) Methods for preparing synthetic freshwaters. *Wat. Res.* 36, 1286-1296.

L115. Flushing and preincubating overnight have an impact on the presence of other gases (N2O, CO2 etc) as well as on the availability of labile organic matter, both dissolved and particulate. A comment on these limitations of sediment slurries should be added in my opinion.

Reply: We are not sure whether purging will significantly affect the DOC pool, which consists mostly of longer carbon-chain molecules rather than volatiles. But certainly, the interference with in situ conditions in general is non-negligible. For example, larger aggregates in the sediments might have been disrupted altering microbe-particle interactions. Also, the pH may have been changed slightly. We will mention this aspect in the revised manuscript.

L117. No information on how much nitrate was added is given. How does it compare to the natural concentrations? If the concentrations used are not saturating for the enzymes in question, relative rates can be dependent on the enzymatic kinetics of each process.

Reply: This information will be included in the revised manuscript. The final $NO_3^-$ conc. in slurry was $120 \pm 2$ µmol $L^{-1}$. The experimental concentration was thus higher than the in situ one and significantly higher than the half saturation concentrations for either of the nitrate-reducing processes.

L118. Details should be added on how the duration of the experiments and time points were determined. Was it based on the oxygen evolution? On some previous experiments? How did the authors know in advance that the incubation should last approx 10 h?

Reply: This information will be added in the revised manuscript. Preliminary tests were performed in order to assess the minimal incubation time required to obtain a measurable signal for $^{15}N-N_2$ measurement through mass spectrometry. Initially, the incubation lasted for 24 h, however, the precise monitoring of $O_2$ conc. over such a long time was quite difficult and $NO_3^-$ conc. dropped quite rapidly, which may have affected the partitioning between the two processes. Therefore, we tested several incubation periods (15 h, 10 h), and ultimately decided to go with 10 h incubations. This way, we were able to obtain clear and robust $^{15}N-N_2$ signals and control of $O_2$ conc. in slurries in parallel experiments was easier to manage.

L121. Any particular reason for the difference in the compensation method? If transfer of liquids was performed under normal atmosphere how did the authors account for any oxygen contamination? How were samples poisoned to prevent microbial activity?

Reply: We decided to use 2 different compensation methods for T0 and other time points in order to avoid, as much as possible, the dilution of the residual nutrient and gas pools by milli-Q and He addition, respectively. At T0, He was used because we collected a greater liquid volume (6 mL) in order to measure nutrient concentrations and DNRA rates, while at T1 and T2, only a 2 mL gas sample was taken in exchange with anoxic milli-Q instead of He. At T3, the incubation was stopped and we did not need to compensate for pressure changes inside the vials anymore, therefore no liquid/gas was added.

As mentioned above, all liquid and gas samples were performed in a glovebox under $N_2$ atmosphere to prevent $O_2$ contamination, and additional contamination of $N_2$ was prevented by flushing the syringe several times with He prior to sampling.

Samples were not poisoned. Liquid samples were filtered (0.2 µm) and stored at -20°C.

L125. Monitoring of O2 concentrations was possible with the optode spots. It is important to know the degree of the "marked decline" was observed and have an estimate of the possible effect on the rates (by looking at the corresponding N species concentrations).

Reply: A table showing the exact $O_2$ concentration measured in slurry at different time points during the incubation will be included in the supplementary information. Oxygen concentration did never drop below 85% of the initial targeted $O_2$ concentration. We will therefore replace "marked decline" by "decline" in the revised manuscript.

L132. Gordon reference is missing.

Reply: Thanks, it will be added.

L140. Did the authors measure organic carbon content which is an important factor in the partitioning between DN and DNRA?

Reply: We did not measure the organic carbon (OC) content during these incubation experiments, but we do not think that changes in OC content over the incubation time played an important role in regulating the partitioning between the two processes in this particular case. In general, the most important factor to consider is the ratio $OC/NO_3^-$, where DNRA is favored at high $OC/NO_3^-$ (e.g. $NO_3^-$-limiting) ratios, and denitrification at lower $OC/NO_3^-$ ratios (e.g. OC-limiting; van den Berg et al., 2016). We agree that changes in OC content over the incubation time likely varied among the different $O_2$ treatments (e.g. higher OM remineralization rates at higher $O_2$ conc.), however, OC concentration measurement in slurry incubation experiments within the frame of another study using sediments from the same setting displayed relatively high OC concentrations (initial OC concentration $\geq 510$ mg/L; Cojean et al., in prep.), suggesting that OC was never limiting over the 10 h incubation period ($8°C$). Similarly, $NO_3^-$ was always present in excess during the entire incubation experiment (maximum decrease in $NO_3^-$ conc. of about $8.1 \pm 1.5$ µM from T0 to Tend). Hence, as neither $NO_3^-$ nor OC were limiting, it is likely that the partitioning between denitrification and DNRA in the different $O_2$ treatments, as compared to control incubations, was not so much affected by changes in the $OC/NO_3^-$ but rather by $O_2$.

van den Berg E.M., Boleij M., Kuenen J.G., Kleerebezem R. and van Loosdrecht M.C.M. (2016) DNRA and denitrification coexist over a broad range of acetate/N- $NO_3^-$ ratios in a chemostat enrichment culture.

L147. Rates were determined based on initial and final data? or regression over time?

Reply: Rates were determined through linear regression of $^{15}N$-$N_2$ concentrations versus time. We will mention it in the revised manuscript.

L162. Maybe I missed it somewhere but I did not find any statistical analysis of the data.

Reply: We will include results of the statistical analyses in the revised manuscript.

The profiles do indeed indicate that the surface sediment of the Figino is less at the surface than further below. Any reason why? However, the profiles of nitrate do not indicate directly that the sediment acts as a sink given that nitrate and nitrite microprofiles occur same as for oxygen at a mm scale.

Reply: Less what? It is unclear to us what the reviewer refers to. Anyway, the sediments represent a strong sink for $NO_x$. Nitrate is consumed quantitatively within the first mm, and there is not the slightest indication for net production of $NO_x$, e.g. indicated by a local $NO_3^-$ maximum.

L180. A treatment with ATU to block nitrification would have clarified many of the issues mentioned here.

Reply: We agree that this would have been a good idea. Nevertheless, we believe that our data set is very much meaningful.

L185 & table 1. Rates of 1.1 umol N g d are very low and probably close to the detection limit of the methods used. Data on detection limits should be added in M&M. This is critical when standard error of 6 replicates is 130% of the mean value.

Reply: The detection of the method is on the order of 0.02 $\mu$mol L$^{-1}$. We will mention this in the M&M section.

L192. The authors could take advantage of their data and provide kinetic rate data calculations to allow for easier comparisons with other studies. Also given the big dispersion of the data it would also provide more quantitative information on the differences between sites and processes. Error of percentages should also be provided L196.

Reply: Given the interference with in situ condition, and the addition of substrates in excess, we do not think that the absolute rates, which must be considered potential rates, matter in the context of this study. We specifically look at the relative rates, relative to controls, and relative with regards to the ratio of denitrification and DNRA. As mentioned above, we will provide information on the error of the percentage numbers.

L214. reduced should be consumed. There is a strong discrepancy as the authors mention between the budget and the order of N species transformations based on stable isotopes and total rates.

Reply: We will replace "reduced" with "consumed".

L217 authors claim that biotic immobilization can be an explanation, however, previously they mention that N species exchangeable fraction was insignificant.

Reply: We referred to the abiotic exchangeable fraction through $NH_4^+$ adsorption to the sediment, which we consider negligible. Yet, we cannot exclude completely the biotic immobilization of $NO_3^-$ in $NO_3^-$-storing microorganisms, especially since we observed the presence of *Beggiatoa* sp., which are able to store significant amounts of $NO_3^-$ in the studied sediments (Cojean et al., in prep.).

L224. Authors should try to put into quantitative perspective their findings given that they have oxygen concentration data for bottom water.

Reply: It is tempting to extrapolate our experimental results to the situation in the water column, and thus assess, for example, expected changes in the denitrification/DNRA ratio with changing $O_2$ concentrations in the near-bottom waters. While there will be a more or less direct link between the bottom water $O_2$ concentrations and the $O_2$ penetration within sediments, we can only speculate about what the $O_2$ concentrations will be right at and below the sediment-water-interface over the annual cycle. In any case, independent of the $O_2$ in bottom waters, $O_2$ will be consumed to completion by aerobic respiration and the oxidation of reduced substances within the first mm of the sediment column.

L237. The effect of oxygen concentration on the partitioning between nitrate reduction will only have an impact on a um scale.

Reply: See last comment. We agree that $O_2$ will only regulate the partitioning between the two nitrate reduction processes in the surficial sediments. Upscaling it to the entire lake-basin is problematic.

L280. Author should show the time evolution of oxygen in the slurries to prove no microniches were created in the slurries. Just mentioning that it is unlikely they were formed is speculation. Where was the spot placed? How many per bottle? Was it always under water? Was any sort of beads were included in the bottles?

Reply: We will upload a table with the $O_2$ conc. measured over time in the supplementary information. Oxygen spots were placed on the glass wall of each serum bottle and the spot remained under water during measurement. No beads were included, but we shook the bottles manually every 30 minutes, prior to $O_2$ measurement, and between $O_2$ readings, the bottles were placed on a shaker (80 rpm). As mentioned above, we did not observe any dramatic decline of $O_2$ conc. in any of the experiments and as the slurries were strongly diluted, we think that microniches did not form.

L285. This makes no sense as pelagic communities are more easily exposed to high oxygen concentrations and thus should be more tolerant, rather than the opposite.

Reply: Good point, and we agree that this is speculation. We will delete the statement.

L292. These sections (4.3-4.5) are purely based on speculations and are not based on any data from this study. Actually, these sections just highlight the limitations of the study.

Reply: We do not agree. In 4.3, for example, we provide conclusive experimental data that indicate that, in relative terms, DNRA is favored over denitrification under less reducing conditions. Going one step further, true, we speculate about the reasons for our observational data, comparing our experimental data with observations from other natural environments. This way we test plausibilities.

These sections are part of an honest scientific discussion, where we try to explain the observations we made. Yes, there is a certain degree of speculation involved, and we cannot always provide final answers. Fair enough. But speculative parts are clearly highlighted as such, and it seems appropriate to discuss the potential implications of our results, even if we cannot provide conclusive evidence for all the statements.

We agree that our study has limitations, and we do not hide them. And these limitations may become most evident in these speculative sections the reviewer refers to. But we opt to leave them in, maybe rephrased even more cautiously, as they are intended to stimulate future work that will verify/falsify the putative regulating mechanisms proposed.

L314. How did the authors ensure lack of H2S in the slurries forming? is is by eliminating SO4 presence or its formation somehow?

Reply: We measured free $H_2S$ concentrations and they were below or close to the detection limit. We will mention this point in the revised manuscript.

L320. Authors have no data to discuss the partitioning between the activity of NAP and NAR enzymes.

Reply: See comment above. Yes, this is speculative, and we will rephrase this part more carefully, so that it is clear that we speculate. But, in the attempt to explain the results, we find, there is some room for speculation. We observe clear experimental evidence for direct $O_2$ control of denitrification versus DNRA. The most obvious "corner" to look for plausible explanation would be at the enzymatic level, and not mentioning this (even if we cannot provide molecular data) would seem unmindful to us. The differential response of denitrifiers and nitrate ammonifiers at least suggests a distinct $O_2$ sensitivity of

the nitrate reductase enzymes involved. Unfortunately, we do not have information on the activity of NAP versus NAR, but in line with what we mentioned above, we would like to propose plausible hypothesis that can then be tested by future work.

L412. The data do not differentiate in any way between NAR and NAP.

Reply: Agreed. Since we do not have any solid data in this regard we will delete "(NAR versus NAP)". But we would like to stick to our more general conclusion that the balance between DNRA and denitrification is modulated by $O_2$ at the nitrate-reducing enzyme level.

Fig 2. Are data single measurements or mean of several replicates? Oxygen in water column of panel b is missing. Y axis scale is not "oxygen penetration depth" but just depth, same as in the first y axis. In the fully mixed lake conditions, oxygen concentrations does not increase above 150 uM yet in the microprofiles O2 is 250 uM.

Reply: The results show the mean downcore concentrations measured in duplicate and triplicate cores for N species and $O_2$ conc., respectively. The second x-axis label is indeed depth at a different scale, and we will correct it. Concentrations measured in bottom water and overlying water during $O_2$ microprofiling do not match because the microprofiling was not performed under in situ conditions but under aerated conditions (and higher temperature). Thus, the $O_2$ profiles reflect the potential biological oxygen demand of the two sampling stations, and the ex-situ $O_2$ penetration depth was meant to serve the biogeochemical characterization of the sediments.

Fig 3. I thin it is better to show the rate data of the same site in the same panel as in Fig 4. Also use different colours to be able to differentiate between thte two.

Reply: We would like to keep the original graph. Indeed, we aimed to show that, independently of the studied site, we observed similar patterns regarding the absolute $O_2$ control on denitrification and DNRA rates. We will, nevertheless, use different colors to differentiate between the two processes.

Fig 4. Mean values can be represented as points with their appropriate error bars plus a statistical analysis to confirm significant differences.

Reply: The statistical results will be included in the manuscript and adequately presented in the figures.

**Response to comments by Anonymous Referee #2**

General comments: In my classical (and perhaps narrow-minded?) view the conclusion on aerobic nitrate reduction and on the partition between DNRA and denitrification in relation to oxygen is still a bit controversial. Therefore I would like the authors to present more information about the actual conditions of their experiments, and apply stronger statistics, as well as present the data from their statistical analyses (it is mentioned that t-tests were used, but no results (p values etc) are presented.

Reply: We thank the anonymous reviewer for her/his insightful comments and questions, which we will address point-by-point below. Indeed, we will, present more information on the experimental conditions and we will include results of the statistical analyses in the revised manuscript.

1. The experiments were performed as slurry incubations in serum bottles to which 15no3 were added. What was the resulting 15NO3 concentration in the slurries?

Reply: The final $NO_3^-$ conc. in slurry was $120 \pm 2$ µmol $L^{-1}$. This information will be included in the revised manuscript.

2.Oxygen was added to the headspace in the slurries and monitored with Oxygen Sensor Spots during the course of the experiments. Please present the data on oxygen concentrations in the serum bottles, trough out the course of the experiment. Is the oxygen concentration constant during the course of the experiment for all treatments or does the concentrations drops to critical levels in some of those? Present those data eventually in the supplementary information.

Reply: The data on $O_2$ concentration over the incubation time will be included in the supplementary information.

Yes, targeted $O_2$ concentrations were as much as possible maintained during the course of the experiments by injection of pure $O_2$ whenever necessary. The measured $O_2$ concentration never dropped below 85 % of the initial targeted concentration in all vials.

3.The rates of DNRA and denitrification were calculated from the accumulation of 15N2 and 15NH4 over the course of an incubation period of 10 hours. Please present data that shows how the concentration of the isotopes changed during the course of the experiment. Do you see a linear increase in the concentration of the isotopes as function of time during the entire incubation period? Can you document such an increase eventually through linear regression analysis? Point 2 and 3 are mandatory for a reliable interpretation of the data. The optimal situation is a) The oxygen concentration does not drop significantly (or not critically ) during the course of the experiment. b) There is a linear increase in the concentration of the isotopes as function of time. An eventual derivation from this situation can compromise the validity of the experiment and the conclusions that can be drawn from the data.

Reply: A representative, exemplary graph showing the evolution of $^{15}N-N_2$ conc. over time will be included in the supplementary information. Essentially in all incubations the concentration of produced $^{15}N-N_2$ exhibited a linear increase over the incubation time (4 time points), which was confirmed by the linear regression analysis (mean of $R^2 > 0.86$ for all experiments). However, concerning the production of $^{15}NH_4^+$, the concentration of $^{15}NH_4^+$ was unfortunately measured only at the beginning and at the end of the incubation only (to avoid dilution during sampling of liquid samples). Yet, preliminary tests of the method with 4 time points displayed a linear increase also of $^{15}NH_4^+$ over time and we can thus expect a similar response in all experiments.

 We agree with the reviewer that point 2 and 3 are very important for a reliable interpretation. But as stated above, a) the $O_2$ concentration did not drop that much below target levels and b) at least for the $^{15}N-N_2$ production, but most likely also for $^{15}N-NH^{4+}$, linear behavior without any significant lack phase or obvious changes in the transformation rates can be confirmed.

4.Statistics is a strong tool for getting sound scientific statements. It is mentioned that t-tests were used to test for differences DNRA and denitrification rates, but no data from these tests are shown. Pleases report those statistical data (eventually in table 1). I would also recommend the use of a statistical method that investigate if the partition of DNRA and denitrification, differs significantly at different oxygen concentrations. Use eventually an ANOVA analysis. You might e.g. use DNRA/(DNRA+Denitrification) i.e. the proportion of DNRA to the measured nitrate reduction rate, as test variable. I understand that the experiment was performed, with replicates and that both rates of denitrification and DNRA were measured simultaneously in the same serum bottle. So it should be possible do a sound statistical analysis. Such an analysis can only strength your interpretation and conclusions. Alternatively use a correlation analysis, where you investigate for significant (positive/negative) correlation, between e.g. DNRA/(DNRA+Denitrification) and the oxygen concentration.

Reply: We thank the reviewer for his valuable inputs regarding the statistical analysis. The t-test results showing the significant difference between denitrification/DNRA rates in the different $O_2$ treatments versus those in the control experiments will be included in the revised manuscript. And yes, we also performed a correlation analysis between the relative contribution of DNRA to the total $NO_3^-$ reduction (%) and the increase of $O_2$ concentration, and the results displayed a positive correlation coefficient of 0.6 and 0.9 for Figino and Melide, respectively. We will include those results in the text of the revised manuscript.

5. A comment to the statement l.185. It is stated that the high background of $14NH_4$ prevent the $15NH_4$ from becoming nitrified, and that the isotope derived rate of DNRA, therefore is not underestimated due to nitrification. I do not think that this argument is valid. The problem is the same as for other tracer studies like S35 based studies of sulfate reduction or 14C based studies of e.g. methane turnover, where you have production and consumption occurring simultaneously. Moeslund et al. (1994) showed with experimental data that Sulfate reduction rates as measured with radiotracers, added to the experimental system at very low concentrations was underestimated if sulfide oxidation was present. Xiao et al. (2018) showed from modeling of a tracer study that the degree of underestimation of rates of methane production was proportional to the incubation period in systems with methane production and methane oxidation. I suggest therefore that you delete this statement. Note that if your overall conclusions regarding DNRA and denitrification and oxygen are correct, an eventual underestimation of DNRA at high oxygen concentrations, would not compromise that conclusion. Moeslund, L., B. Thamdrup, and B. Barker Jørgensen. 1994. Sulfur and iron cycling in a coastal sediment: Radiotracer studies and seasonal dynamics. Biogeochemistry 27: 129-152. Xiao, K.-Q., F. Beulig, H. Røy, B. B. Jørgensen, and N. Risgaard-Petersen. 2018. Methylotrophic methanogenesis fuels cryptic methane cycling in marine surface sediment. Limnol. Oceanogr. 63: 1519-1527.

R5: We thank the reviewer for her/his valuable input and the literature provided. In the revised manuscript, we will delete the statement.

**List of relevant changes**

**Adjustments according to our responses to comments by anonymous referee #1**

Line 86: "ventilated" changed to "oxygenated" (Ref. 1, R1)
Line 101: Information about the temperature added (Ref.1, R2)
Line 103: Additional information about the core sectioning procedure included (Ref. 1, R3)
Line 108: The reference for the artificial lake water included (Ref. 1, R4)
Line 113-115: Additional information about the potential influence of He purging on the in situ conditions provided in the Materials and Methods section 4.3 (Ref. 1, R5)
Line 116: Details provided about the $NO_3^-$ added in slurry experiments (in the Materials and Methods section 4.3) (Ref. 1, R6)
Line 118-120: Additional information concerning the design of the experimental set-up (e.g. duration of the experiments) added in the Materials and Methods section 2.3 (Ref. 1, R7)
Line 139 & 285: A table showing the $O_2$ concentrations during the incubation experiments added in the Supplementary Information Table SI.1 (Ref.1, R8)
Line 145: Additional details about the detection limit of the $NO_x$-analyser included in the Materials and Methods section 2.5 (Ref. 1, R9)
Line 153: Additional information concerning the calculation of $^{15}N$-$N_2$ production rates added in the Materials and Methods section 2.6 (Ref. 1, R10)
Line 195-202: Additional information about the error of the percentage numbers provided (Ref. 1, R11)
Line 214: We replaced "reduced" with "consumed (Ref. 1, R12)
Line 289: Statement about the greater resistance of benthic microbial communities than pelagic ones at high $O_2$ concentration deleted (Ref. 1, R13).
Line 321: Additional information concerning the $H_2S$ concentrations in slurries provided (Ref. 1, R14)
Line 321/325/330/350/353: Parts of the discussion that were quite speculative rephrased (Ref. 1, R15)
Line 422: Statement about "Nar versus Nap" deleted (Ref. 1, R16)

Figure 3: Statistical analysis of the data included in the figure (Ref. 1, R17)
References/Figures: several minor adjustments of graphs according to the referee's remarks

**Adjustments according to our responses to comments by anonymous referee #2**

Line 116: Detail about the $NO_3^-$ amendments provided in the Materials and Methods section 4.3 (Ref. 2, R1)
Line 139 & 285: A table showing the $O_2$ concentration trends during the incubation experiments added in the Supplementary Information Table SI.1 (Ref.2, R2)
Line 154: An exemplary graph showing the evolution of $^{15}N$-$N_2$ production over incubation time (and regression analyses) for two $O_2$ levels included in the supplementary information (Ref. 2, R3)
Figure 3 & line 203: Statistical analysis of the data included in Figure 3, in the results section 3.3 and in the Supplementary Information Fig. SI.2 (Ref. 2, R4)
Line 187: Statement suggesting that, due to high $^{14}NH_4^+$ background, the portion of $^{15}NH_4^+$ being nitrified was negligible, deleted (Ref. 2, R5)

[revised manuscript text omitted]

Cojean, A.N.Y., *et al.*, Biogeosciences, 2019

*Correspondence*: adeline.cojean@unine.ch

**Table SI.1: Average $O_2$ concentrations in replicate slurries (see Table 1) at different time points during the incubation period (~ 10 h). The first time point (T1) was always at 30 minutes after the beginning of the incubation while the other time points (T2, T3, …, T9) were different for each treatment (i.e. more frequent $O_2$ measurements at low $O_2$ levels).**

**Commenté [MOU1]:** Ref. 1, R8; Ref. 2, R2

| | Average $O_2$ concentration during incubation ($\mu$mol $L^{-1}$) | $O_2$ concentration in slurry at different time points ($\mu$mol $L^{-1}$) | | | | | | | | |
|---|---|---|---|---|---|---|---|---|---|---|
| | | T1 | T2 | T3 | T4 | T5 | T6 | T7 | T8 | T9 |
| **Figino** | 1.2 | 1.4 | 1.2 | 1.4 | 1.2 | 1.2 | 1.1 | 1.0 | 1.1 | 1.2 |
| | 2.0 | 1.7 | 2.2 | 2.0 | 2.2 | 1.9 | 2.0 | 1.7 | 2.0 | 2.1 |
| | 2.8 | 2.2 | 2.6 | 2.9 | 3.1 | 2.9 | 2.9 | 2.9 | 2.8 | 3.0 |
| | 3.4 | 3.0 | 3.2 | 3.9 | 3.2 | 3.6 | 3.7 | | | |
| | 4.1 | 3.5 | 3.5 | 3.9 | 4.3 | 4.1 | 4.2 | 4.9 | | |
| | 8.6 | 7.5 | 7.8 | 8.8 | 9.1 | 8.1 | 8.9 | 10.1 | | |
| | 16.0 | 17.6 | 17.1 | 15.0 | 16.7 | 16.5 | 13.6 | 15.5 | 15.8 | |
| | 24.1 | 26.3 | 25.8 | 22.9 | 24.5 | 21.0 | 24.2 | | | |
| | 38.0 | 39.3 | 35.2 | 39.5 | 36.8 | 38.3 | | | | |
| | 61.3 | 66.8 | 62.1 | 57.2 | 56.9 | 63.4 | | | | |
| | 78.6 | 81.0 | 84.8 | 73.1 | 81.6 | 73.6 | 77.4 | | | |
| **Melide** | 0.8 | 0.8 | 0.8 | 0.6 | 0.8 | 0.8 | 0.7 | 1.0 | 1.1 | |
| | 1.8 | 1.2 | 2.0 | 1.7 | 2.0 | 1.8 | 1.7 | 1.9 | 2.1 | |
| | 2.9 | 2.7 | 3.0 | 2.9 | 3.1 | 2.8 | 3.0 | | | |
| | 3.8 | 3.2 | 3.5 | 3.3 | 3.8 | 3.7 | 4.1 | 3.5 | 4.3 | 4.4 |
| | 4.9 | 4.8 | 4.8 | 5.1 | 4.7 | 4.7 | 4.9 | 5.2 | | |
| | 9.0 | 8.3 | 8.2 | 8.8 | 8.7 | 9.5 | 9.4 | 10.3 | 11.1 | 11.1 |
| | 13.1 | 11.3 | 11.3 | 13.3 | 12.7 | 13.8 | 13.8 | 15.1 | | |
| | 21.3 | 20.6 | 19.0 | 21.4 | 21.4 | 24.2 | | | | |
| | 44.4 | 40.3 | 36.4 | 47.8 | 46.7 | 50.6 | | | | |
| | 58.6 | 44.6 | 53.3 | 56.3 | 52.0 | 57.8 | 66.6 | 68.9 | 69.0 | |
| | 73.2 | 39.3 | 70.8 | 73.8 | 69.7 | 76.8 | 81.2 | 86.4 | 87.3 | |

**Table SI.2: Results of the t-test analysis of the difference between denitrification and DNRA rates in $O_2$ treatments and those in control experiments. *P* values with significant differences ($P < 0.05$) are highlighted in bold character.**

Commenté [MOU2]: Ref. 2, R4

| | $O_2$ concentration in slurry ($\mu$mol $L^{-1}$) | Denitrification | DNRA |
|---|---|---|---|
| **Figino** | 1.2 | 0.3529 | **0.0023** |
| | 2.0 | **0.0011** | **0.0127** |
| | 2.8 | **0.0003** | 0.2137 |
| | 3.4 | **0.0004** | **0.0305** |
| | 4.1 | **0.0001** | **0.0182** |
| | 8.6 | **< 0.0001** | **0.0003** |
| | 16.0 | **0.0056** | **< 0.0001** |
| | 24.1 | **< 0.0001** | **0.0093** |
| | 38.0 | **< 0.0001** | **< 0.0001** |
| | 61.3 | **< 0.0001** | **0.0002** |
| | 78.6 | **< 0.0001** | **< 0.0001** |
| **Melide** | 0.8 | **0.0099** | **0.0256** |
| | 1.8 | **0.0002** | **0.0008** |
| | 2.9 | **0.0055** | **0.0019** |
| | 3.8 | **0.0001** | **0.0002** |
| | 4.9 | **0.0045** | **0.0002** |
| | 9.0 | **< 0.0001** | **< 0.0001** |
| | 13.1 | **< 0.0001** | **0.0459** |
| | 21.3 | **< 0.0001** | **0.0005** |
| | 44.4 | **< 0.0001** | **0.0433** |
| | 58.6 | **< 0.0001** | **0.0062** |
| | 73.2 | **< 0.0001** | **< 0.0001** |

[Figure]

**Figure SI.1:** **Exemplary times-series of $^{30}N_2$ concentration data during incubation experiments in two different $O_2$ treatments (triplicate) at Melide.**

**Commenté [A3]:** Ref. 2, R3

[Figure]

**Figure SI.2: Relative contribution of DNRA (%) to total $NO_3^-$ reduction at variable $O_2$ concentrations at Figino and Melide.**

---

## Author Response (AR2)

The authors have done a good job incorporating new information and clarifying aspects of the paper.

I only have a couple of corrections/suggestions.

We thank the reviewer for her/his valuable inputs. We will make changes in the revised manuscript accordingly.

L109. If procedures from Line 106-109 were made in anaerobic glove box as you mention in the reply, this should be included in the text.

Reply: This will be included in the revised manuscript.

L115. I did not argue tha purging will affect the DOC pool. Purging can potentially affect the balance of gases (CO2, H2, CH4) and anaerobic processes in the sediment. Pre-incubation may affect the availability of labile DOC (even without purging).

Reply: We thank the reviewer for clarifying her/his point.

L121. Please mention that you filtered the samples in the glove box before freezing.

Reply: This will be included in the revised manuscript.

L173. In my previous comment "The profiles do indeed indicate that the surface sediment of the Figino is less at the surface than further below. Any reason why? However, the profiles of nitrate do not indicate directly that the sediment acts as a sink given that nitrate and nitrite microprofiles occur same as for oxygen at a mm scale." I did not express myself correctly.

Your profiles of NO3 are in cm, not mm as you mention in your answer.

With a resolution of 1 cm slices, the authors would not be able to notice the nitrification/denitrification zone which can be as little as 1-2mm to 1 cm. So it may be that by integrating porewater profiles at such coarse resolution can indicate the wrong conclusion. What would happen if I make a porewater profile of oxygen in a highly productive sediment at a 1 cm resolution? I would see a decrease from 200 to 0 and I would assume that it is consuming oxygen whereas in reality it will be releasing O2 across the sediment water interface. Same here. Without core incubations, you cannot really know if the sediment is a net sink or source.

My other question was, why do the authors think it might be that the sediment at Figino seems less reactive near the surface than further below? Changes in porosity? changes in OC?

Reply: We thank the reviewer for clarifying her/his point, and indeed we made a mistake in our previous answer, where we meant to write cm, and not mm. We agree that the depth resolution of the $NO_3^-$ profiles is not sufficient to resolve the exact porewater nitrate concentration gradient at the sediment-water interface (SWI). This is why we mentioned in the text that the sediments represent a sink for $NO_3^-$ within the top centimeter of the sediments and did not draw any conclusion about the $NO_3^-$ turnover within the first millimeters. Given the multi-fold core-incubation evidence (e.g., Wenk et al. 2014) for the sediments taking up nitrate, it would be brave to assume that there might be a hidden $NO_3^-$ concentration peak just below the SWI, which we were not able to resolve. Nevertheless, we do not question that $NO_3^-$ is produced via nitrification in the 1-2 mm of the sediments, yet it was entirely consumed within the top centimeter, yielding a net concentration gradient that points to a flux of nitrate into the sediments.

This may be a misunderstanding. We do not think that the sediments at Figino are less reactive near the surface than further below. But there is biogeochemical evidence (e.g., oxygen penetration depth) that the sediments at Figino are less reactive than those at Melide, which may indeed be due to a higher reactivity of the sedimentary organic matter at Melide.

Wenk, C. B., Zopfi, J., Gardner, W. S., McCarthy, M. J., Niemann, H., Veronesi, M. and Lehmann, M. F.: Partitioning between benthic and pelagic nitrate reduction in the Lake Lugano south basin, Limnol. Oceanogr., 59(4), 1421–1433, doi:10.4319/lo.2014.59.4.1421, 2014.

**List of relevant changes**

**Adjustments according to our responses to comments by anonymous referee #1**

Line 107: Additional information mentioning that incubations were performed into an anaerobic chamber was included in the text (Ref.1, R1)

Line 126: Details about the processing (e.g. filtering and freezing) of the samples were included in the text (Ref.1, R2)

[revised manuscript text omitted]